# Enhanced machine learning and hybrid ensemble approaches for Coronary Heart Disease prediction

**Maurice Wanyonyi**[1]*, **Zakayo Ndiku Morris**[1],
**Faith Mueni Musyoka**[2], **Dominic Makaa Kitavi**[1,3]

**1** Department of Mathematics and Statistics, University of Embu, Embu, Kenya, **2** Department of Computing and Information Technology, University of Embu, Embu, Kenya, **3** College of Computer Studies, De La Salle University, Manila, Philippines

☯ These authors contributed equally to this work.
* mauricewanyonyi27@gmail.com

## Abstract

Coronary heart disease (CHD) remains the leading cause of mortality worldwide, disproportionately affecting low- and middle-income countries where diagnostic resources are limited. Traditional statistical models often fail to deliver adequate predictive accuracy in complex, high-dimensional, and imbalanced health datasets. To develop and evaluate enhanced machine learning and hybrid ensemble models for the prediction of coronary heart disease, with a focus on improving diagnostic performance, interpretability, and applicability in resource-constrained settings. We utilized a nationally representative dataset of 253,680 individuals from the Behavioral Risk Factor Surveillance System. Preprocessing included normalization and balancing via the Synthetic Minority Oversampling Technique (SMOTE). Baseline models—Decision Trees, Random Forests, Gradient Boosting, and Support Vector Machines—were compared against improved versions: Adaptive Noise–Resistant Decision Tree (ADNRT), Hybrid Imbalanced Random Forest (HIRF), Pruned Gradient Boosting Machine (PGBM), and Enhanced Support Vector Machine (ESVM). Ensemble approaches (stacking, boosting, bagging, Bayesian model averaging and majority voting) were implemented and evaluated using accuracy, sensitivity, specificity, and area under the curve (AUC). Calibration and learning curves were also analyzed. Enhanced models consistently outperformed their baseline counterparts. PGBM achieved the highest sensitivity (90.8%), while HIRF demonstrated the best overall calibration and balance (AUC = 0.937; sensitivity = 88.4%; specificity = 82.9%). The stacking ensemble emerged as the best-performing model with an accuracy of 87.2%, sensitivity of 89.6%, specificity of 84.7%, and AUC of 0.94. Calibration and learning curve analyses confirmed strong generalizability and low overfitting across ensemble models. Hybrid ensemble machine learning models significantly outperform traditional classifiers in CHD prediction, offering high accuracy, robustness, and

**Data availability statement:** The data underlying the results presented in the study are available from IEEE DataPort (https://ieee-dataport.org/documents/heart-disease-dataset). The dataset is publicly accessible and was retrieved online. All data used in this study are secondary and were not collected by the authors. The data can be made available upon subscription from IEEE DataPort, or alternatively, provided by the authors upon reasonable request.

**Funding:** The author(s) received no specific funding for this work.

**Competing interests:** The authors have declared that no competing interests exist.

interpretability. These models present a scalable framework for implementing AI-driven diagnostic tools in low–resource environments, potentially transforming early detection and prevention of coronary heart disease.

## Introduction

The most widespread type of cardiovascular disease (CVD), coronary heart disease (CHD), has been the primary cause of death globally since it kills more than 7 million people each year and accounts for around 85 per cent of all deaths associated with cardiovascular issues [1,2]. Though much progress has been made in the field of cardiovascular care, the incidence of CHD is increasing, especially in low- and middle-income countries (LMICs), where there is still limited access to services that can help prevent and diagnose CHD at an early stage and treat it adequately [3,4]. Demands for effective, scalable, and legible tools that enable early prediction and intervention are more emergent now than ever. In sub-Saharan Africa (SSA), CHD is becoming more prevalent among younger, economically productive groups, further increasing the public health and economic consequences [5,6]. CHD has reached epidemic proportions in Kenya, leading to an increase in the number of hospital admissions and heart failure, which have been worsened by risk factors, including hypertension, obesity, diabetes, and HIV-related complications [7,8].

Early CHD mortality risk prediction is of immense importance in reducing mortality and improving patient outcomes, especially in situations with resource constraints. Logistic regression and Cox proportional hazards models are among the statistical models traditionally used to identify primary cardiovascular disease (CHD) risk factors and predict outcomes [9,10]. These models are appreciated because they are interpretable and straightforward; however, in high-dimensional health data situations, they tend to misrepresent the complex and nonlinear relationships that exist therein. Over the past few years, extensive research has been conducted on machine learning (ML) methods due to their ability to learn from data, detect previously unknown patterns, and achieve improved predictive performance [11,12]. The random forest, support vector machines, gradient boosting, and multi-layer perceptron algorithms have shown promising outcomes in CHD prediction [13–15].

Nevertheless, there are a few limitations to machine learning (ML) models. Most of them face problems of overfitting, lack of transparency, and inadequate generalization when used in different populations, particularly in low- and middle-income countries (LMICs), where the quality of data and its completeness may not be improved [16,17]. The issue of imbalanced datasets is not unique to CHD. [18] emphasized that class imbalance severely affects model performance across domains such as credit risk, fraud detection, and medical diagnosis. Their work with enhanced sparse autoencoders shows how feature learning can mitigate imbalance and improve classifier performance. In addition, although a wide variety of research has been conducted to develop ML-based diagnostic technologies, there is still a distinct research gap in models aimed at the predictive progression of CHD, an indispensable consideration when it comes to the application of preventative and long-term management of disease [19,20]. Moreover, ML models have been commonly assessed in isolation,

without being fully integrated and compared with classical statistical methods; their contextual applicability and interpretability in clinical decision-making are inevitably constrained.

To bridge this gap in methodology, the ensemble learning approach has emerged as an influential strategy that integrates the strength of single models in order to reduce their deficiencies. Different ensemble methods, including bagging, boosting, and stacking, have been shown to significantly outperform predictive models across diverse clinical settings [21–23]. These techniques attempt to fuse predictions of many base learners, where the base learners are usually a combination of interpretable models, like logistic regression, and strong learners, like random forest and XGBoost, to deliver robustness, better performance, and generalization. Specifically, stacked generalization has proven to achieve better performance in predicting CHD because it learns to assign the best possible weights to the predictions of base models, minimizing both bias and variance at the same time [14,24].

Current literature underscores the advantages of ensemble approaches in cardiovascular risk modeling. For instance, [21] proposed an improved ensemble method based on CART models, achieving accuracy scores of over 93% on benchmark datasets. [23] developed a stacked ensemble model that outperformed standalone classifiers, reporting an accuracy of 92.34%. Likewise, [14] validated ensemble methods that utilize feature selection and hyperparameter tuning to achieve robust classification performance. [25] demonstrated that filter-based feature selection combined with cost-sensitive AdaBoost significantly improved the early detection of chronic kidney disease, achieving near-perfect classification. In the cardiac domain, [26] developed a decision support system for predicting mortality in cardiac patients, leveraging SMOTE for imbalance correction and a $\chi^2$-based feature ranking combined with optimized random forests, achieving 94.59% accuracy with only 10 features. This highlights the importance of integrating feature selection with classifier optimization in medical diagnosis. Meanwhile, [24] found that ensemble models outperformed traditional base classifiers by an average of 1.96% in accuracy. These findings collectively suggest that integrating multiple predictive paradigms can significantly enhance diagnostic power and mitigate the limitations of individual classifiers.

Nevertheless, the majority of these ensemble strategies have been tested on high-income datasets and often lack contextual relevance to LMICs such as Kenya. Most models are designed with little consideration for resource constraints, data sparsity, and infrastructural limitations commonly encountered in LMIC healthcare systems [27,28]. Furthermore, limited research has explored how ensemble models perform when classical statistical models–typically favored for their interpretability—are integrated with complex ML models in a structured hybrid framework. This integration could potentially balance the trade-off between prediction accuracy and interpretability, a crucial requirement in healthcare settings where clinicians must trust and understand model outputs.

Recent advances in heart disease prediction have incorporated explainable AI (XAI), hybrid learning frameworks, and generalizable deep models. Recent research has also integrated enhanced sparse autoencoders with Softmax regression to achieve high predictive accuracy in diseases such as CKD, cervical cancer, and heart disease [29]. Comparable advances are evident in stroke prediction, where an intelligent learning system combining autoencoders with linear discriminant analysis (LDA) achieved 99.24% accuracy and balanced performance across sensitivity and specificity [30]. These works further underscore the value of combining unsupervised feature learning with robust classifiers for improved performance in imbalanced healthcare datasets. For instance, CardioRiskNet [31] combines attention mechanisms, active learning, and XAI to deliver highly accurate and interpretable risk predictions yet focuses primarily on deep architectures and does not benchmark against traditional or ensemble-based learners.

The HXAI-ML framework [32] integrates advanced resampling strategies and SHAP/LIME/PIA explanations, achieving superior results on benchmark datasets but without clustering or ensemble fusion layers. Related efforts have also been applied in other health domains, such as hepatitis B diagnosis, where SHAP-based interpretability helped clinicians identify bilirubin as the most critical feature influencing mortality [33]. In addition, [34] demonstrate that boosting algorithms like XGBoost and CatBoost perform well in CHD prediction, identifying key clinical features with support from conformal classifiers, though their work does not explore hybrid ensemble construction or subgroup analysis. More recently, stroke prediction has benefited from class balancing and data augmentation, where a CBDA-ResNet50 model achieved a balanced

accuracy of 98.27%, highlighting the effectiveness of combining deep architectures with imbalance-handling strategies [35]. Meanwhile, [36] propose a transparent prediction interface tailored for clinicians using XAI and user-centric design but with limited performance benchmarking across multiple learner types.

In this study, we seek to address existing methodological and contextual gaps in CHD prediction by developing a hybrid modeling approach that integrates diverse machine learning algorithms into robust ensemble frameworks. Specifically, our study pursues two primary objectives: (1) to evaluate the predictive performance of individual machine learning models using a large, population-based dataset; and (2) to develop ensemble models that combine these algorithms to enhance both prediction accuracy and interpretability.

To ensure robustness and relevance, the study employs a comprehensive methodological framework, including pre-processing (handling missing data, class imbalance, and outliers), diagnostic statistics, feature selection, and hyperparameter tuning through grid search. The individual models explored include decision trees, random forests, support vector machines, and gradient boosting machines. These are evaluated using standard metrics such as accuracy, precision, recall, F1-score, and area under the receiver operating characteristic curve (AUC–ROC). This multi-layered strategy aims not only to improve prediction performance but also to provide interpretable insights that can be acted upon in real–world clinical settings.

In contrast to recent studies–such as CardioRiskNet [31], HXAI-ML [32], and other explainable AI frameworks [34,36], our work addresses three key gaps. (1) The development of enhanced machine learning variants—including Adaptive Noise Resistant Decision Trees (ANRDT) and Hybrid Imbalanced Random Forests (HIRF)—specifically tailored for CHD prediction in resource-constrained environments; (2) A comprehensive benchmarking of ensemble strategies, including advanced techniques such as Bayesian Model Averaging, to assess predictive performance and generalization across models; and (3) The integration of unsupervised clustering analysis to identify latent CHD risk subgroups, offering both predictive enhancement and clinically meaningful subgroup interpretation. Collectively, these contributions advance the methodological landscape of CHD prediction by improving accuracy, interpretability, and contextual relevance especially within low- and middle-income country (LMIC) settings such as Kenya. Beyond healthcare, similar methods have been applied in financial risk prediction, where stacked sparse autoencoders improved credit card default prediction [37]. These cross-domain successes illustrate the versatility of ensemble and feature-learning approaches, reinforcing their potential for CHD prediction in resource-limited settings.

Accordingly, this work provides a structured comparison between individual machine learning models, their enhanced variants, and hybrid ensemble frameworks. Unlike prior studies that either focus solely on standalone models or lack contextual relevance to LMICs, this study emphasizes methodological rigor and practical utility. By benchmarking these models on a standardized dataset and evaluation framework, the research contributes actionable insights for developing scalable, interpretable, and accurate CHD prediction systems to support early detection and improved clinical decision-making in resource-limited settings.

## Materials and methods

### Data source

Coronary heart disease dataset was sourced from the IEEE Dataport repository, a well–established platform for sharing high-quality datasets in health research, including epidemiological and cardiovascular disease studies [38]. The selected dataset (the Behavioral Risk Factor Surveillance System (BRFSS) dataset) was collected by the U.S. Centers for Disease Control and Prevention (CDC) and includes responses from 253,680 individuals, with 22 variables encompassing demographic, behavioral and lifestyle, clinical information, and healthcare access & general well-being. Out of the total records, 229,787 do not indicate CHD, while 23,893 are classified as CHD-positive based on self-reports.

The decision to use BRFSS dataset is driven by both its comprehensive nature and the relevance of its variables to CHD prediction in diverse populations. Despite being U.S.–based, BRFSS captures risk factors (e.g., smoking, cholesterol

levels, blood pressure, diabetes) that are also prevalent in LMICs. Given the limited availability of large-scale, population-level CHD datasets in LMICs like Kenya, BRFSS serves as a valuable data source for model training. The model can later be fine-tuned and validated using smaller, country-specific datasets from LMICs to ensure adaptability and accuracy in local settings.

## Preprocessing

Missing values in the data was addressed using appropriate imputation techniques, such as multiple imputation or k–nearest neighbors (KNN), depending on the nature of the missing data pattern. The methods used to detect outliers included the interquartile range (IQR) and Mahalanobis distance. The presence of noise was eliminated using noise reduction techniques such as smoothing filters, feature correlation analysis, and variance thresholding. Lastly, normalization and standardization were employed to guarantee uniform scaling of features and enhance the model training performance.

## Handling class imbalance

In the original data, the CHD–positive cases class was extremely low in relative values to the total sample (less than 10 per cent of the total sample). This is why Synthetic Minority Oversampling Tecnique (SMOTE) was employed. SMOTE creates synthetic samples of the minority class (CHD positive) by interpolation among existing minority observations in the instance (feature) space. In contrast to random oversampling, SMOTE stabilizes the risk of overfitting because it adds variability to the synthetic samples, thus resulting in a more balanced and generalizable decision boundary for the classifiers [39,40]. Subsequently, SMOTE applied to the dataset returned a balanced dataset, which was used to develop a model since the classifier became more sensitive to CHD cases. At the same time, its specificity remained unaltered. SMOTE is a popular method used to address class imbalance in machine learning datasets by generating synthetic examples of the minority class. The basic mathematical equation behind SMOTE is as follows:

Let:

- $x_i$ be a minority class sample (feature vector),
- $x_{nn}$ be one of its $k$-nearest neighbors (also from the minority class),
- $\lambda$ be a random number in the range [0,1],

Then a synthetic sample $x_{new}$ is generated as:

$$x_{new} = x_i + \lambda \cdot (x_{nn} - x_i) \tag{1}$$

## Sample size determination

The original number of observations in the present research was 253,680. Following the balance between the classes using SMOTE, the size of the whole sample was significantly expanded to N = 459,574. This larger dataset presented a great computational challenge as it had to be balanced with a better representation. When trained on this complete dataset, machine learning algorithms required too many iterations to contend with and usually caused the system to crash or become unresponsive due to insufficient memory. To mitigate this, the Yamane formula was used to calculate the representative subsample, since it is a simplification of calculating the sample size of a given population and the level of precision that is desired [41,42]. The formula is expressed as:

$$n = \frac{N}{1 + Ne^2} \tag{2}$$

Where:

- $n$ is the calculated sample size,
- $N$ is the population size,
- $e$ is the margin of error (level of precision).

For this study:

$$N = 459{,}574, \quad e = 0.01$$

Substituting into the formula:

$$
\begin{aligned}
n &= \frac{459{,}574}{1 + 459{,}574 \times (0.01)^2} \\
&= \frac{459{,}574}{1 + 459{,}574 \times 0.0001} \\
&= \frac{459{,}574}{1 + 45.9574} \\
&= \frac{459{,}574}{46.9574} \\
&\approx 9{,}789
\end{aligned}
\tag{3}
$$

Therefore, the ultimate sample size chosen to train the machine learning models was about n = 9,789 observations. In order to make sure that this sample was representative enough of the structure and diversity of the whole dataset, especially in terms of class imbalance and essential demographic or clinical sub-populations, the stratified sampling method was used. Stratified sampling involves dividing the populations into uniform subunits or strata (e.g., by classes or labels, age intervals, or gender), and each of these strata is sampled proportionately [43,44]. The method proves particularly useful within healthcare-driven data sets where minorities (i.e., disease outcomes) are poorly represented [45]. Stratified sampling increases representativeness and makes all significant subpopulations well-represented in the sample. It is primarily suggested to enhance the results of training machine learning models to eliminate bias and variance in unbalanced datasets [46].

## Base model selection

Several base machine learning models were considered in this study and include Decision Trees, Random Forests, Gradient Boosting Machines, and Support Vector Machines. The reason for selecting these models is their performance with complex and high-dimensional data, as well as their popularity among predictive modelling problems. Nevertheless, there are inherent limitations within each of these models that may contravene its performance, as noted in recent research. To mitigate these difficulties, better versions of the models have been put forward to make them more robust and more predictive.

**Decision Trees** are simple yet effective decision models that divide the data by performing feature thresholds and dividing the processed data into zones, each of which is associated with a specific value of the target. They are highly interpretable but also increase the risk of overfitting and are very sensitive to noise [17]. To enhance noise immunity and overfitting, this study utilized an Adaptive Noise-Resistant Decision Tree (ANRDT), which is an improved form of decision tree. In contrast to traditional trees, which apply hard threshold splits, ANRDT also applies probabilistic (soft) splits via a sigmoid function so that smooth decisions on the boundaries of the decisions can be made. By doing so, the model will be less reliant on low-level input data variances. Moreover, model complexity and generalization are regulated with pruning

and regularization techniques, which help ANRDT to be more stable and tolerant of noise in real-world data. The formula for the improved decision tree is as follows:

$$f(x) = \frac{1}{1 + e^{\beta(x-\theta)}}$$

(4)

Where $f(x)$ is the predicted probability, $\beta$ is the steepness of the decision boundary, $x$ is the predictor value, and $\theta$ is the split threshold.

**Random Forests** increase the stability and performance of a model by averaging out multiple predictions made by different decision trees. Nevertheless, they can have problems with skewed data and cannot give precise estimations of feature importance [47]. This study proposed a Hybrid Imbalanced Random Forest (HIRF) to solve the issues of class imbalance and interpretability associated with the standard Random Forests. HIRF improves on the performance by setting the SMOTE to equalize the underrepresented classes and make the models fair. Also, the SHapley Additive exPlanations (SHAP) is employed to assign meaningful scores to individual tree based on their contributions to the prediction and thus to enhance the interpretability of the feature importance. The weighted ensemble method, therefore, focuses on trees that extract important patterns in the data and produce predictions that are both more accurate and explainable.The HIRF model is given by

$$\hat{y} = \sum_{i=1}^{n} w_i T_i(x)$$

(5)

Where $\hat{y}$ is the predicted target, $n$ is the number of trees, $T_i(x)$ is the prediction from the $i$-th tree, and $w_i$ is the weight assigned to each tree based on SHAP values.

**Gradient Boosting Machines:** iteratively construct decision trees, refining predictions by minimizing errors and focusing on residuals. While highly effective, GBMs are susceptible to overfitting and sensitive to outliers [48]. To mitigate these issues, this study implemented a Pruned Gradient Boosting Machine (PGBM), which integrates early stopping and Huber loss to enhance model robustness and generalization. The Huber loss function is defined as:

$$L(y, \hat{y}) = \begin{cases} \frac{1}{2}(y - \hat{y})^2, & \text{if } |y - \hat{y}| \leq \delta \\ \delta\left(|y - \hat{y}| - \frac{\delta}{2}\right), & \text{if } |y - \hat{y}| > \delta \end{cases}$$

(6)

In this equation, $\delta$ is the threshold for identifying outliers, $y$ is the true value, and $\hat{y}$ is the predicted value.

Early Stopping monitors validation performance and stops training when further iterations no longer improve the model, preventing overfitting. Huber Loss Function combines Mean Squared Error (MSE) and Mean Absolute Error (MAE) to reduce the influence of extreme outliers while maintaining smooth gradient updates.

**Support Vector Machines (SVM):** are powerful supervised learning classifiers that construct a maximum-margin hyperplane to optimally separate data points of different classes. Despite their effectiveness, standard SVMs are often criticized for being difficult to interpret and for requiring meticulous hyperparameter tuning [16].

To address these limitations, this study introduced an Enhanced Support Vector Machine (ESVM) framework. The ESVM improves interpretability by incorporating SHapley Additive exPlanation (SHAP) values, which attribute the contribution of each feature to individual predictions. Additionally, the model employs Bayesian optimization to automatically tune hyperparameters, thereby improving efficiency and predictive performance.

The standard SVM seeks to find the optimal hyperplane by solving the following convex optimization problem:

$$\min_{w,b,\xi} \frac{1}{2}\|w\|^2 + C\sum_{i=1}^{n} \xi_i \quad \text{subject to} \quad y_i(w^T x_i + b) \geq 1 - \xi_i, \quad \xi_i \geq 0$$

(7)

To enable non-linear classification, the ESVM utilizes the Radial Basis Function (RBF) kernel, which maps input features into a higher-dimensional space. The RBF kernel is defined as:

$$K(x, x') = \frac{1}{e^{\gamma \|x - x'\|^2}} \tag{8}$$

Where:

- $K(x, x')$ is the kernel function measuring similarity between input vectors $x$ and $x'$,
- $\|x - x'\|^2$ is the squared Euclidean distance,
- $\gamma$ is a hyperparameter controlling the kernel's smoothness.

The dual optimization problem, often used for kernel-based SVMs, is formulated as:

$$\max_{\alpha} \sum_{i=1}^{n} \alpha_i - \frac{1}{2} \sum_{i,j=1}^{n} \alpha_i \alpha_j y_i y_j K(x_i, x_j)$$

$$\text{subject to} \quad 0 \leq \alpha_i \leq C, \quad \sum_{i=1}^{n} \alpha_i y_i = 0 \tag{9}$$

The resulting decision function used to classify new instances is given by:

$$f(x) = \text{sign}\left(\sum_{i=1}^{n} \alpha_i y_i K(x_i, x) + b\right) \tag{10}$$

To enhance model transparency, the ESVM uses SHAP values to decompose the prediction:

$$f(x) = \phi_0 + \sum_{i=1}^{M} \phi_i \tag{11}$$

Where:

- $f(x)$ is the predicted output,
- $\phi_0$ is the base value (expected prediction),
- $\phi_i$ is the SHAP value for feature $i$,
- $M$ is the total number of input features.

This combination of SHAP-based interpretability and Bayesian hyperparameter tuning makes ESVM a robust and transparent model suitable for high-stakes domains such as healthcare prediction.

## Data split and model training

In order to facilitate the adequate development and assessment of the model, the BRFSS dataset has been split into three subsets: 70% as training, 15% as validation and 15% as a test set. Such stratified split was able to create effective models without the danger of overfitting and guarantee the user-friendly applicability of the findings [49]. The base models were fitted on the training set, and the validation set was utilized to help fine–tune the hyperparameters and to help tune the model performance. The test set was not used in training and validation but instead was used as an independent test

set to test the predictive power of the final model. Further improvement of the reliability was achieved by using a 5-fold cross-validation of the training phase. Here, the training data was split into five equal phases or "folds". Each time there is an iteration, one fold should be kept as a temporary validation set, and the rest of the folds are utilized to train the model. This process is repeated five times, with each fold used as the validation set once. Cross–validation not only supports robust model tuning but also helps mitigate overfitting by ensuring that the model performs consistently across different subsets of the data. Model training aimed to identify the classifier that best predicts CHD outcomes based on the selected predictor variables while minimizing the loss function on the test set (Fig 1).

## Clustering analysis

To explore hidden subpopulations among individuals affected by coronary heart disease (CHD), an unsupervised clustering approach was applied independently across four domains: demographics, clinical characteristics, behavioral and lifestyle patterns, and healthcare access, combined with general well-being. A fifth integrative clustering was then performed using all variables to capture cross-domain profiles. The K–Means algorithm was selected for its efficiency and suitability for structured epidemiological data [50], partitioning the dataset into $K$ clusters by minimizing the sum of squared distances between data points and their respective centroids, as described by the objective function:

$$J = \sum_{i=1}^{k} \sum_{x_j \in C_i} \|x_j - \mu_i\|^2 \tag{12}$$

where $\mu_i$ denotes the centroid of cluster $i$, and $C_i$ represents the set of data points assigned to that cluster. prior to clustering, all continuous features were standardized using Z-score normalization, while categorical variables were appropriately encoded to ensure balanced influence across the algorithm. Domain-specific clustering enabled focused analysis of key

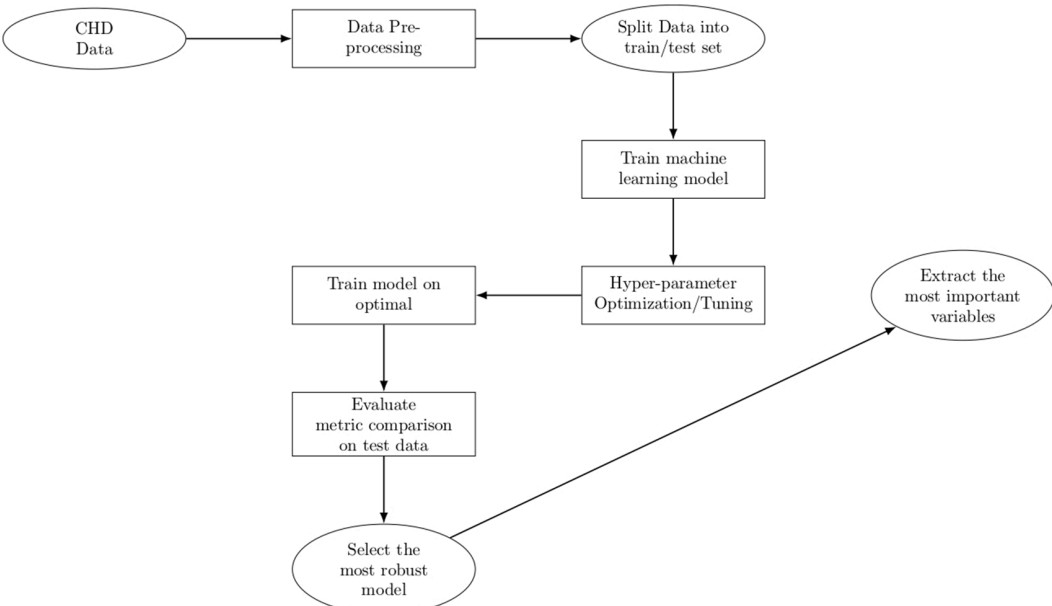

**Fig 1**. **Machine learning framework.** The figure illustrates the overall machine learning framework used in this study, including data preprocessing, model training, evaluation, and ensemble methods.

determinants: demographic attributes (e.g., age, sex, education, income), clinical indicators (e.g., blood pressure, cholesterol, diabetes), behavioral risk factors (e.g., smoking, diet, alcohol use), and access/well-being measures (e.g., insurance, general health, mental and physical health days). Optimal cluster numbers were identified using silhouette scores. To evaluate associations with CHD outcomes, Chi-square tests, and multivariable logistic regressions were conducted, with cluster labels as predictors. This multi-level strategy revealed nuanced patient subgroups, enhancing the understanding of how combined social, clinical, and behavioral factors shape CHD risk and guiding more precise cardiovascular prevention strategies.

## Ensemble learning

To enhance the predictive performance of the ensemble framework, this study integrated multiple ensemble learning techniques: Boosting, Bagging, Stacking, Majority Voting, and Bayesian Model Averaging (BMA). Each technique was implemented and compared to determine the one that achieves the highest predictive accuracy and reliability for CHD.

**Boosting:** is an ensemble learning technique that aims to convert a set of weak learners into a strong learner through iterative refinement. Unlike bagging, where models are trained independently and in parallel, boosting trains models sequentially. Each new model is trained to correct the errors made by the previous ensemble of learners by giving more focus (weight) to the misclassified observations. Formally, given a training set

$$\{(x_i, y_i)\}_{i=1}^n \quad \text{where} \quad x_i \in \mathbb{R}^d, \quad y_i \in \{-1, +1\} \tag{13}$$

for binary classification, boosting builds an additive model of the form:

$$F(x) = \sum_{m=1}^{M} \alpha_m h_m(x) \tag{14}$$

where:

- $h_m(x)$ is a weak learner (e.g., a shallow decision tree) at iteration $m$,
- $\alpha_m$ is the weight assigned to learner $h_m$,
- $F(x)$ is the final strong classifier after $M$ iterations.

**AdaBoost:** In AdaBoost (Adaptive Boosting), the weights of misclassified samples are increased so that the next learner focuses more on difficult cases. The model minimizes the exponential loss:

$$\mathcal{L} = \sum_{i=1}^{n} \exp(-y_i F(x_i)) \tag{15}$$

At each step:

1. The weak learner $h_m(x)$ is trained to minimize the weighted classification error:

$$\epsilon_m = \sum_{i=1}^{n} w_i^{(m)} \cdot \mathbb{I}(h_m(x_i) \neq y_i) \tag{16}$$

2. The model weight $\alpha_m$ is computed as:

$$\alpha_m = \frac{1}{2} \ln \left( \frac{1 - \epsilon_m}{\epsilon_m} \right) \tag{17}$$

3. The sample weights are updated:

$$w_i^{(m+1)} = w_i^{(m)} \cdot \exp(-\alpha_m y_i h_m(x_i)) \tag{18}$$

and then normalized to sum to 1.

**Gradient Boosting Machines:** Gradient Boosting generalizes boosting by optimizing a differentiable loss function $\mathcal{L}(y, F(x))$ using gradient descent. At each iteration, a new learner $h_m(x)$ is fitted to the negative gradient of the loss with respect to the current model prediction:

$$g_i^{(m)} = - \left[ \frac{\partial \mathcal{L}(y_i, F(x_i))}{\partial F(x_i)} \right]_{F(x) = F_{m-1}(x)} \tag{19}$$

Then $h_m(x)$ is trained to predict $g_i^{(m)}$, and the model is updated as:

$$F_m(x) = F_{m-1}(x) + \alpha_m h_m(x) \tag{20}$$

where $\alpha_m$ is the learning rate (step size). Popular implementations of boosting algorithms include XGBoost, LightGBM, and CatBoost, which introduce improvements in speed, scalability, regularization, and support for categorical features. These optimizations make gradient boosting one of the most powerful techniques in modern machine learning.

**Bagging (Bootstrap Aggregating):** is an ensemble learning technique that aims to reduce model variance and prevent overfitting by training multiple models on different bootstrap samples (random samples with replacement) of the original dataset. Each individual model, often referred to as a *base learner*, is trained independently, and their predictions are aggregated to produce the final output. For regression tasks, the final prediction is the average of the individual predictions:

$$\hat{y} = \frac{1}{M} \sum_{m=1}^{M} h_m(x) \tag{21}$$

where $h_m(x)$ is the prediction from the $m$-th model and $M$ is the total number of models. For classification tasks, majority voting is typically used:

$$\hat{y} = \text{mode} \{h_1(x), h_2(x), \ldots, h_M(x)\} \tag{22}$$

A widely–used bagging-based algorithm is **Random Forest**, which constructs an ensemble of decision trees. In addition to training on bootstrap samples, Random Forest introduces further randomness by selecting a random subset of features at each split in the tree-building process. This decorrelates the individual trees and further improves ensemble diversity, stability, and predictive accuracy.

**Stacking:** is a hierarchical ensemble learning technique that combines the predictions of multiple base models through a meta-model (also known as a *level-1 model*). Unlike bagging and boosting, which use homogeneous models and direct aggregation (e.g., averaging or voting), stacking allows heterogeneous base learners and learns how to best combine their predictions. In the first stage, $M$ base models $h_1(x), h_2(x), \ldots, h_M(x)$ are trained on the training data. The predictions of these models on a hold-out validation set are collected to form a new dataset:

$$Z = \{(h_1(x_i), h_2(x_i), \ldots, h_M(x_i))\}_{i=1}^{n} \tag{23}$$

A meta-model $H(Z)$ is then trained on this new dataset $Z$, learning to combine the outputs of the base models. The final prediction is:

$$\hat{y} = H(h_1(x), h_2(x), \ldots, h_M(x)) \tag{24}$$

Stacking can be used for both classification and regression. In classification, the meta-model is often a logistic regression or support vector machine; in regression, it could be a linear regressor or gradient boosting machine.

**Majority Voting and Averaging:** are simpler ensemble strategies where no meta-model is involved. For classification, the final prediction is made based on the majority class predicted by the base models:

$$\hat{y} = \text{mode}\{h_1(x), h_2(x), \ldots, h_M(x)\} \tag{25}$$

For regression, the predictions are averaged:

$$\hat{y} = \frac{1}{M} \sum_{m=1}^{M} h_m(x) \tag{26}$$

While majority voting and averaging improve robustness through consensus, stacking often yields better performance by learning complex interactions among model predictions.

**Bayesian Model Averaging (BMA):** is an ensemble learning technique that incorporates model uncertainty by averaging over multiple models, each weighted by its posterior probability given the data. Rather than selecting a single "best" model, BMA considers the contribution of all candidate models in making predictions, thus improving robustness and accuracy.

The final prediction is computed as a weighted sum of the individual model predictions:

$$\hat{y} = \sum_{i=1}^{M} p_i f_i(x) \tag{27}$$

where:

- $f_i(x)$ is the prediction from the $i$-th base model,
- $p_i = P(\mathcal{M}_i \mid D)$ is the posterior probability of model $\mathcal{M}_i$ given the observed data $D$,
- $M$ is the total number of models,
- $\hat{y}$ is the final ensemble prediction.

The posterior probabilities satisfy the normalization constraint:

$$\sum_{i=1}^{M} p_i = 1 \tag{28}$$

The posterior probability $p_i$ for each model $\mathcal{M}_i$ can be computed using Bayes' theorem:

$$p_i = \frac{P(D \mid \mathcal{M}_i) \, P(\mathcal{M}_i)}{\sum_{j=1}^{M} P(D \mid \mathcal{M}_j) \, P(\mathcal{M}_j)} \tag{29}$$

where $P(D \mid \mathcal{M}_i)$ is the marginal likelihood (model evidence) and $P(\mathcal{M}_i)$ is the prior probability of model $\mathcal{M}_i$. BMA combines the strengths of multiple models while explicitly accounting for the uncertainty in model selection. This leads to more calibrated predictions and often outperforms model selection approaches that rely on a single model, especially when model evidence is comparable across candidates.

## Evaluation metrics

The performance of the ensemble methods was compared with the classical statistical models and the individual machine learning models in terms of sensitivity, specificity, accuracy, F1– score, and AUC–ROC. These measures are developed based on the confusion matrix, which is a primary instrument in assessing the classification model performance. It gives a table of actual and predicted classifications, which offers an in-depth analysis of how the model works. Table 1 represents the confusion matrix that corresponds to the classification of CHD cases where the rows indicate the true classes of the CHD cases and the columns indicate the predicted classes of the CHD cases. The matrix is an overview of how well the model can differentiate between true negatives (non–CHD patients correctly predicted), true positives (CHD patients correctly predicted), false positives (non–CHD patients incorrectly classified as CHD) and false negatives (CHD patients not correctly predicted by the model).

Sensitivity (true positive rate) measures the proportion of actual positive cases correctly identified by the model. A sensitivity of 1 signifies perfect identification of all true positives, while a value of 0 indicates that the model failed to identify any true positives.

$$\text{Sensitivity} = \frac{TP}{TP + FN} \tag{30}$$

where $TP$ = True Positives, $FN$ = False Negatives.

Accuracy measures the proportion of correct predictions (both true positives and true negatives) among all predictions made. An accuracy of 1 indicates that the model made correct predictions for every case, while an accuracy of 0 means that every prediction was incorrect.

$$\text{Accuracy} = \frac{TP + TN}{TP + FN + FP + TN} \tag{31}$$

where $TN$ = True Negatives, $FP$ = False Positives.

Specificity (true negative rate) measures the proportion of actual negative cases correctly identified by the model. A specificity of 1 indicates perfect identification of all true negatives, while a value of 0 means the model failed to identify any true negatives.

$$\text{Specificity} = \frac{TN}{TN + FP} \tag{32}$$

where $TP$ = True Positives, $FN$ = False Negatives.

$$\text{F1–score} = \frac{2 \cdot \text{Precision} \cdot \text{Recall}}{\text{Precision} + \text{Recall}} = \frac{2TP}{2TP + FP + FN} \tag{33}$$

**Table 1**. **Confusion matrix structure.**

| | Predicted Positive | Predicted Negative |
|---|---|---|
| Actual Positive | True Positive (TP) | False Negative (FN) |
| Actual Negative | False Positive (FP) | True Negative (TN) |

Table notes: This table outlines the standard structure of a binary classification confusion matrix, indicating the relationships between actual and predicted outcomes.

AUC-ROC (Area Under the Receiver Operating Characteristic curve) evaluates the model's ability to discriminate between positive and negative cases. An AUC of 1 signifies a perfect model with optimal discrimination. A value of 0 means the model is completely ineffective and is classifying oppositely.

### Software and analytical tools

All data cleaning, preprocessing, modeling, and visualization were conducted using **Python version 3.13**. The analytical workflow integrated a suite of specialized libraries to ensure efficiency, accuracy, and reproducibility.

**Data manipulation and preprocessing** were performed using `pandas` and `numpy`, while `scikit-learn` was used for model building, evaluation, and implementation of resampling techniques such as Synthetic Minority Oversampling Technique (SMOTE).

**Ensemble and tree-based models** such as Random Forest, Gradient Boosting, and Stacking were implemented via `scikit-learn`, with enhanced versions (e.g., Pruned GBM and Noise-Resistant Trees) incorporating customized Python classes and hyperparameter tuning using `GridSearchCV` and `RandomizedSearchCV`.

**Data visualization** and exploratory analysis were conducted using `matplotlib`, `seaborn`, and `plotly`, with interactive visualizations used for exploratory cluster analysis and survival curve interpretation. Clustering and dimensionality reduction employed `scikit-learn`'s `KMeans`, and silhouette analysis functions.

All scripts were version-controlled using Git, and `Jupyter Notebooks` were used for iterative experimentation. Reproducibility was further ensured through fixed random seeds, isolated virtual environments (via `venv` or `conda`), and consistent preprocessing pipelines.

### Ethical consideration

Ethical approval was deemed unnecessary for this study despite involving human subjects because the data used was secondary, and there was no physical contact with human subjects during data collection/retrieval. The data can be availed to anyone upon subscription at IEEE*Dataport* (Heart Disease Dataset)

## Results

### Data balancing and class distribution

Initially, there was a significant class imbalance, with only 9.42% of individuals classified as having experienced heart disease or a heart attack (Yes = 1). The vast majority (90.58%) were classified as not having experienced such events (No = 0). This reflects a common issue in medical and epidemiological datasets, where the condition of interest occurs in a relatively small population subset. Such imbalance poses a significant challenge for machine learning models, particularly those relying on classification, as they tend to be biased toward predicting the majority class. This can lead to high overall accuracy but poor sensitivity (recall) for the minority class, which, in this context, is the class of most significant clinical interest. The SMOTE technique was applied to generate synthetic examples of the minority class. As seen in Table 2, the result was a perfectly balanced distribution of 50.0% for each class, thereby ensuring that the classifier receives equal representation of both outcomes during training. This balanced distribution improves the model's ability to learn minority class patterns and predict positive cases of CHD more effectively (Figs 2 and 3).

### Clustering analysis

**Demographic clusters.** The demographic clustering produced eight clusters with a silhouette score of 0.273, indicating moderate cohesion among groups. A chi-square test showed a strong association with CHD ($p<.001$). As shown in Table 3, logistic regression identified Clusters 1 through 3 as high-risk groups, each with odds ratios exceeding 3.75 ($p<.001$). Cluster 6 also presented elevated odds of CHD (OR = 2.03, $p<.001$), while Cluster 5 displayed a modest but

**Table 2. Class distribution of *Heart Disease or Attack* before and after SMOTE.**

| Heart Disease or Attack | Before SMOTE (%) | After SMOTE (%) |
|---|---|---|
| No (0) | 90.58 | 50.00 |
| Yes (1) | 9.42 | 50.00 |

Table notes: This table presents the percentage distribution of the target classes before and after applying the SMOTE technique to balance the dataset.

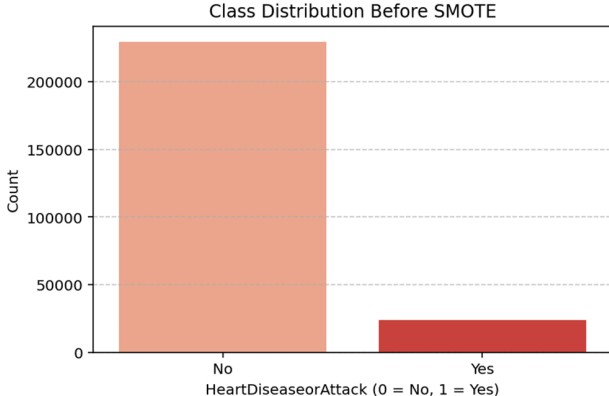

**Fig 2. Class distribution before SMOTE.** The figure shows the distribution of the target classes in the dataset before applying SMOTE. Class 0 (individuals without heart disease or heart attack) is significantly more frequent than class 1 (individuals with the condition), highlighting a pronounced class imbalance with nearly 91% of cases belonging to class 0.

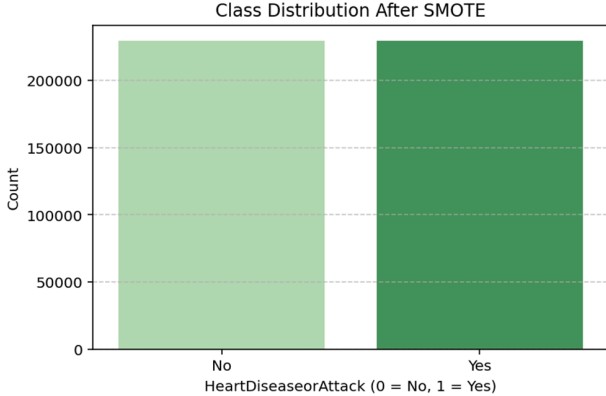

**Fig 3. Class distribution after SMOTE.** The figure illustrates the distribution of the target classes after applying SMOTE. Both class 0 and class 1 are represented equally, each comprising 50% of the dataset, demonstrating the successful balancing of the minority class and the effectiveness of SMOTE in addressing class imbalance.

statistically significant increase in risk (OR = 1.24, $p$ = .027). These patterns suggest that demographic subpopulations characterized by socioeconomic or age-related vulnerabilities are disproportionately affected by CHD (Figs 4–8).

**Clinical clusters.** Clustering based on clinical characteristics produced nine distinct clusters and yielded a high silhouette score of 0.666, indicating strong structural separation between subgroups. A chi-square test confirmed a statistically significant association between cluster membership and CHD status ($p$<.001). As shown in Table 4, Cluster 5 exhibited an exceptionally high risk of CHD with an odds ratio (OR) of 12.37, followed by Cluster 8 (OR = 4.73), Cluster 7 (OR = 2.42),

**Table 3. Logistic regression for demographic clusters in predicting CHD.**

| Cluster | Coef | Std Err | p-value | OR | Lower CI | Upper CI |
|---------|------|---------|---------|-----|----------|----------|
| Intercept | -0.836 | 0.070 | < 0.001 | 0.43 | 0.38 | 0.50 |
| Cluster 1 | 1.332 | 0.088 | < 0.001 | 3.79 | 3.19 | 4.50 |
| Cluster 2 | 1.341 | 0.088 | < 0.001 | 3.82 | 3.21 | 4.54 |
| Cluster 3 | 1.346 | 0.088 | < 0.001 | 3.84 | 3.23 | 4.57 |
| Cluster 4 | 0.628 | 0.089 | < 0.001 | 1.87 | 1.57 | 2.23 |
| Cluster 5 | 0.217 | 0.098 | 0.027 | 1.24 | 1.02 | 1.51 |
| Cluster 6 | 0.706 | 0.094 | < 0.001 | 2.03 | 1.69 | 2.43 |
| Cluster 7 | 0.344 | 0.096 | < 0.001 | 1.41 | 1.17 | 1.70 |

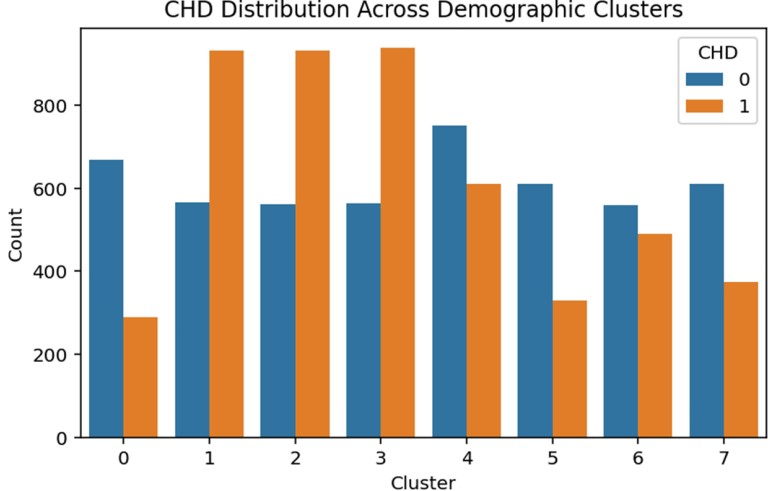

**Fig 4. CHD distribution across demographic clusters.** The figure shows the proportion of coronary heart disease (CHD) cases within each demographic cluster. Each bar represents the percentage of individuals diagnosed with CHD in that cluster, illustrating variability in risk across the population subgroups.

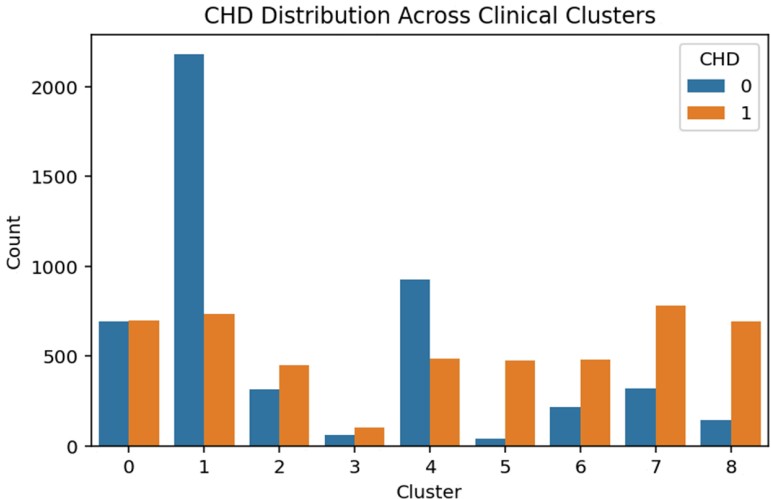

**Fig 5. CHD distribution across clinical clusters.** The figure shows the distribution of coronary heart disease (CHD) cases across clinical clusters. Higher CHD counts are concentrated in clusters associated with elevated odds, supporting the observed variation in risk among different clinical subgroups.

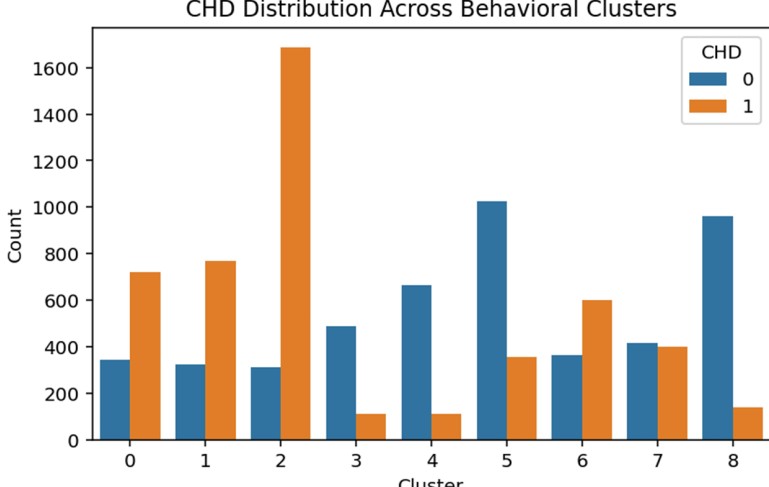

**Fig 6**. **CHD distribution across behavioral clusters.** The figure shows the distribution of coronary heart disease (CHD) cases across behavioral clusters. This visualization highlights how CHD cases vary among different behavioral subgroups, supporting the observed patterns in risk factors.

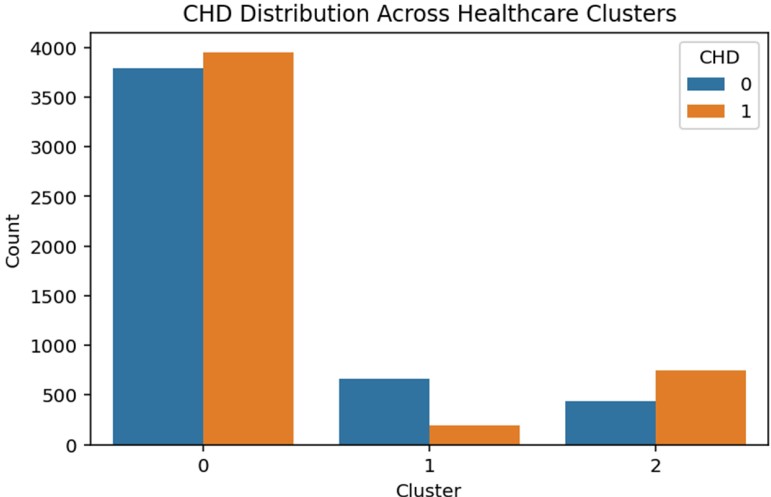

**Fig 7**. **CHD distribution across healthcare clusters.** The figure shows the distribution of coronary heart disease (CHD) cases across healthcare clusters. The visualization highlights variations in CHD burden among different healthcare subgroups, corroborating the trends observed in the study.

and Cluster 6 (OR = 2.17). In contrast, Clusters 1 and 4 showed protective associations (ORs < 1.0), suggesting these subgroups may contain individuals with fewer or better-managed clinical risk factors.

   **Behavioral clusters.** The behavioral clustering resulted in nine distinct groups with a high silhouette score of 0.700, indicating strong internal cohesion and separation between clusters. The chi-square test confirmed a significant association with CHD ($p$<.001). As shown in Table 5, logistic regression revealed that Cluster 2 was associated with significantly elevated odds of CHD (OR = 2.59, $p$<.001). In contrast, Clusters 3, 4, 5, and 8 demonstrated strong protective effects, with odds ratios ranging from 0.07 to 0.17, likely representing behaviorally healthier groups (e.g., non-smokers, physically active individuals). Cluster 1 did not show a statistically significant difference ($p$ = .215), suggesting a more neutral behavior-risk profile.

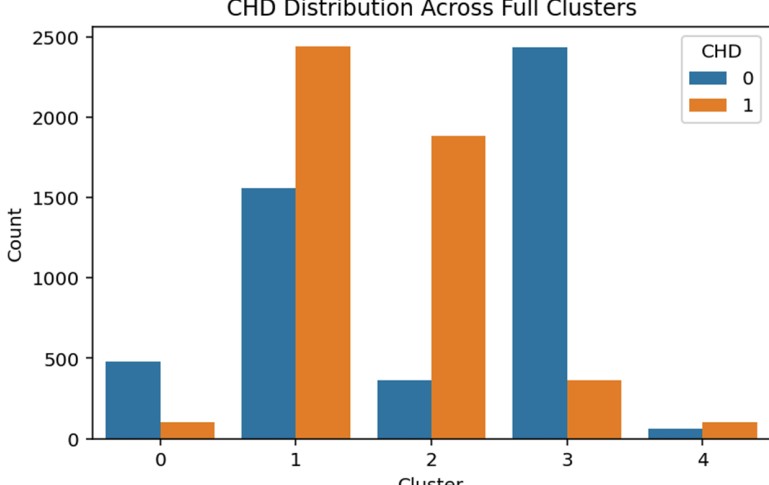

**Fig 8**. **CHD distribution across full feature clusters.** The figure shows the distribution of coronary heart disease (CHD) cases across full feature clusters. This visualization highlights the variation in CHD burden when considering all features, supporting the overall patterns observed across the dataset.

**Table 4**. **Logistic regression for clinical clusters in predicting CHD.**

| Cluster | Coef | Std Err | p-value | OR | Lower CI | Upper CI |
|---|---|---|---|---|---|---|
| Intercept | 0.009 | 0.054 | 0.872 | 1.01 | 0.91 | 1.12 |
| Cluster 1 | -1.096 | 0.069 | < 0.001 | 0.33 | 0.29 | 0.38 |
| Cluster 2 | 0.343 | 0.091 | < 0.001 | 1.41 | 1.18 | 1.68 |
| Cluster 3 | 0.502 | 0.172 | 0.003 | 1.65 | 1.18 | 2.31 |
| Cluster 4 | -0.649 | 0.078 | < 0.001 | 0.52 | 0.45 | 0.61 |
| Cluster 5 | 2.515 | 0.177 | < 0.001 | 12.37 | 8.74 | 17.49 |
| Cluster 6 | 0.777 | 0.098 | < 0.001 | 2.17 | 1.79 | 2.63 |
| Cluster 7 | 0.886 | 0.085 | < 0.001 | 2.42 | 2.05 | 2.87 |
| Cluster 8 | 1.554 | 0.106 | < 0.001 | 4.73 | 3.84 | 5.82 |

**Table 5**. **Logistic regression for behavioral clusters in predicting CHD.**

| Cluster | Coef | Std Err | p-value | OR | Lower CI | Upper CI |
|---|---|---|---|---|---|---|
| Intercept | 0.744 | 0.066 | < 0.001 | 2.11 | 1.85 | 2.39 |
| Cluster 1 | 0.116 | 0.093 | 0.215 | 1.12 | 0.94 | 1.35 |
| Cluster 2 | 0.950 | 0.090 | < 0.001 | 2.59 | 2.17 | 3.09 |
| Cluster 3 | -2.236 | 0.124 | < 0.001 | 0.11 | 0.08 | 0.14 |
| Cluster 4 | -2.524 | 0.121 | < 0.001 | 0.08 | 0.06 | 0.10 |
| Cluster 5 | -1.799 | 0.090 | < 0.001 | 0.17 | 0.14 | 0.20 |
| Cluster 6 | -0.239 | 0.093 | 0.011 | 0.79 | 0.66 | 0.95 |
| Cluster 7 | -0.784 | 0.096 | < 0.001 | 0.46 | 0.38 | 0.55 |
| Cluster 8 | -2.684 | 0.112 | < 0.001 | 0.07 | 0.05 | 0.09 |

**Healthcare and well-being clusters.** Three clusters were identified in the healthcare and well-being domain, with a silhouette score of 0.347. Although the cluster cohesion was lower than in other domains, the association with CHD remained statistically significant ($p<.001$). As shown in Table 6, Cluster 2 exhibited elevated odds of CHD (OR = 1.62, $p<.001$), potentially reflecting individuals facing barriers such as poor access to care, high out-of-pocket costs, or lower self-rated health. In contrast, Cluster 1 showed a strong protective association (OR = 0.28, $p<.001$), likely representing individuals with stable healthcare access and more favorable well-being indicators.

**Table 6.** Logistic regression for healthcare clusters in predicting CHD.

| Cluster | Coef | Std Err | p-value | OR | Lower CI | Upper CI |
|---|---|---|---|---|---|---|
| Intercept | 0.043 | 0.023 | 0.061 | 1.04 | 1.00 | 1.09 |
| Cluster 1 | -1.268 | 0.085 | < 0.001 | 0.28 | 0.24 | 0.33 |
| Cluster 2 | 0.482 | 0.064 | < 0.001 | 1.62 | 1.43 | 1.84 |

**Full feature clusters.** Clustering across the full feature set resulted in five integrated clusters, though the silhouette score was relatively low (0.077), indicating weak separation among clusters. Nevertheless, the association with CHD remained highly significant ($p<.001$). As presented in Table 7, Cluster 2 exhibited exceptionally high odds of CHD (OR = 24.36), followed by Clusters 1 and 4 (ORs = 7.32 and 7.78, respectively). These clusters likely capture individuals with combined demographic, clinical, and behavioral vulnerabilities. In contrast, Cluster 3 had significantly lower odds of CHD (OR = 0.70, $p = .004$), suggesting the presence of a potentially resilient subgroup. Despite the reduced silhouette cohesion, this integrated domain provided valuable insight into the overlapping risk factors influencing CHD.

## Confused matrix of baseline ensemble models

**Bayesian Model Averaging (BMA).** The model's sensitivity is calculated as:

$$\text{Sensitivity} = \frac{876}{876 + 124} \approx 87.6 \tag{34}$$

which shows its ability to correctly identify patients with CHD. The specificity, representing its performance on the negative class, is:

$$\text{Specificity} = \frac{839}{839 + 161} \approx 83.9 \tag{35}$$

These values suggest that while BMA performs reasonably well in identifying CHD-positive individuals, it has a moderate rate of false alarms. This trade-off is expected due to the averaging nature of the method, which might smooth over sharp decision boundaries (Figs 9–13).

**Majority voting.** The sensitivity is given by:

$$\text{Sensitivity} = \frac{873}{873 + 127} \approx 87.3 \tag{36}$$

and the specificity is:

$$\text{Specificity} = \frac{854}{854 + 146} \approx 85.4 \tag{37}$$

**Table 7.** Logistic regression results for full feature clusters predicting coronary heart disease (CHD).

| Cluster | Coef | Std Err | p-value | OR | Lower CI | Upper CI |
|---|---|---|---|---|---|---|
| Intercept | -1.541 | 0.109 | < 0.001 | 0.21 | 0.17 | 0.26 |
| Cluster 1 | 1.991 | 0.113 | < 0.001 | 7.32 | 5.86 | 9.14 |
| Cluster 2 | 3.193 | 0.123 | < 0.001 | 24.36 | 19.15 | 30.99 |
| Cluster 3 | -0.353 | 0.122 | 0.004 | 0.70 | 0.55 | 0.89 |
| Cluster 4 | 2.052 | 0.196 | < 0.001 | 7.78 | 5.30 | 11.43 |

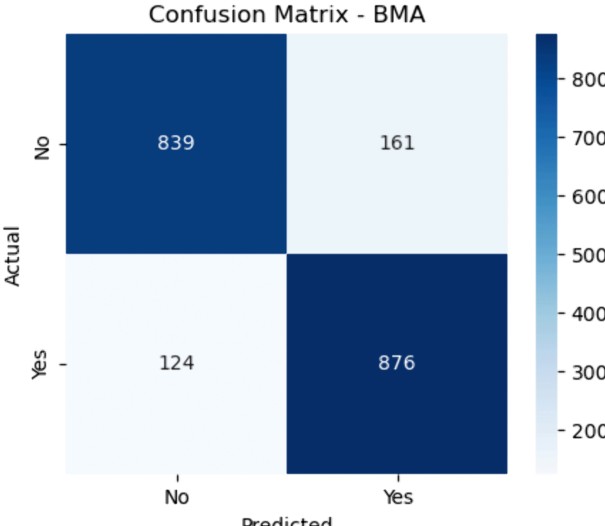

**Fig 9. Confusion matrix for Bayesian Model Averaging (BMA).** The figure shows the confusion matrix for the baseline Bayesian Model Averaging (BMA) model, illustrating the distribution of true positives, true negatives, false positives, and false negatives. This method combines probabilistic predictions from various base classifiers using Bayesian principles to produce an averaged output. The model correctly classified 839 non–CHD cases (true negatives) and 876 CHD-positive cases (true positives). However, it also misclassified 161 nonC–HD instances as CHD (false positives) and failed to detect 124 true CHD cases (false negatives).

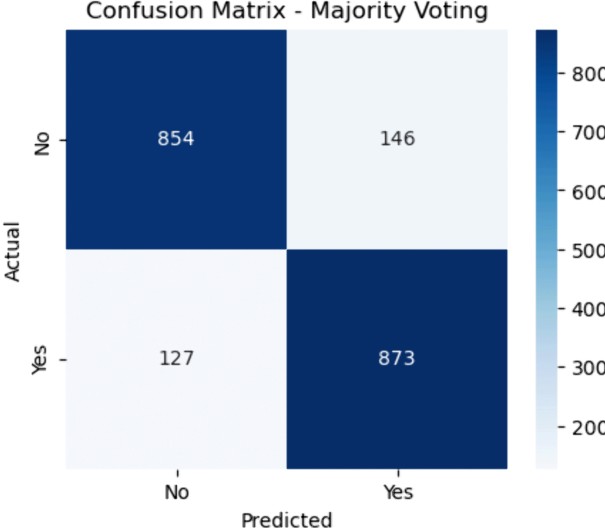

**Fig 10. Confusion matrix for majority voting.** The figure shows the confusion matrix for the baseline Majority Voting ensemble, a straightforward yet robust method where each base learner casts a vote and the majority prediction is selected. The model achieved 854 true negatives and 873 true positives. It also misclassified 146 non-CHD patients as positive (false positives) and missed 127 CHD cases (false negatives).

These results suggest that Majority Voting strikes a decent balance between identifying CHD-positive individuals and minimizing false positives. However, due to the equal weighting of all classifiers, it may overlook the strengths of stronger learners in certain cases.

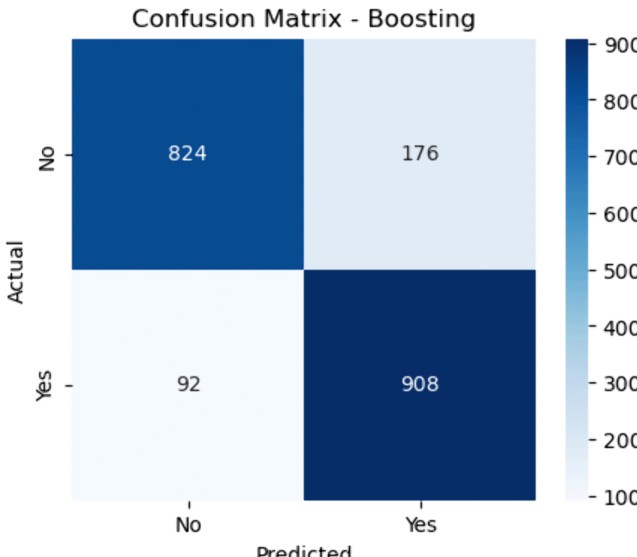

**Fig 11**. **Confusion matrix for boosting (baseline).** The figure shows the confusion matrix for the baseline Boosting model, which trains weak learners sequentially while correcting previous errors. The model correctly identified 908 CHD-positive patients (true positives) and 824 non-CHD individuals (true negatives). However, it misclassified 92 CHD patients (false negatives) and 176 non–CHD patients (false positives).

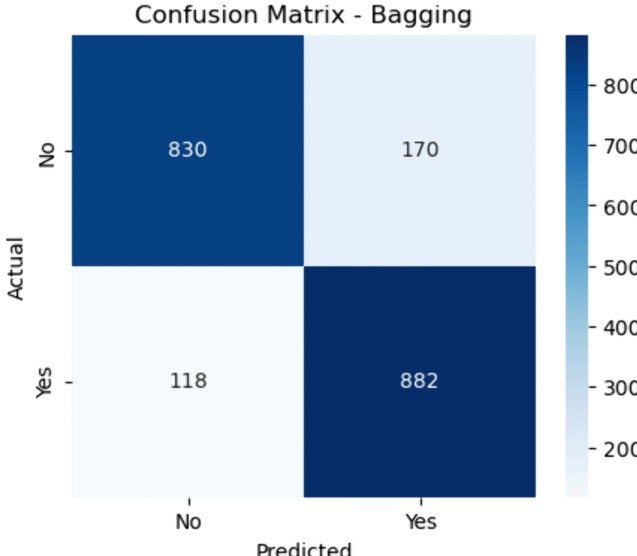

**Fig 12**. **Confusion matrix for bagging (baseline).** The figure shows the confusion matrix for the baseline Bagging model—based on bootstrap aggregation—highlighting 830 true negatives and 882 true positives. It recorded 170 false positives and 118 false negatives.

### 0.0.1 Boosting. Sensitivity is computed as:

$$\text{Sensitivity} = \frac{908}{908 + 92} \approx 90.8 \tag{38}$$

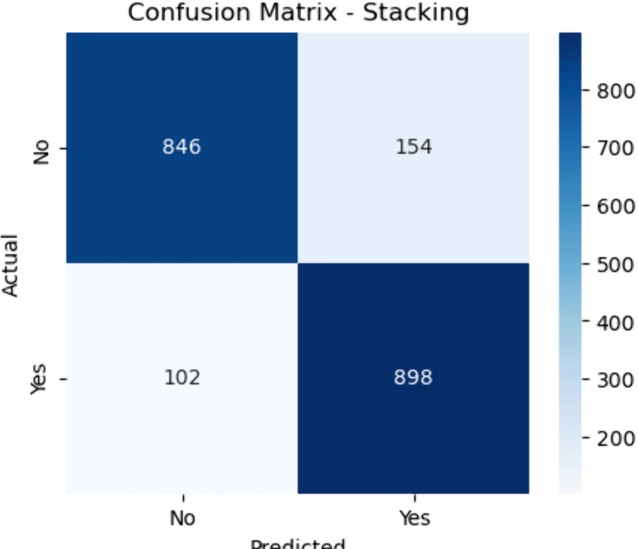

**Fig 13**. **Confusion matrix for stacking (baseline).** The figure shows the confusion matrix for the baseline Stacking ensemble, which uses a meta-learner to combine predictions from multiple base models. The stacking model correctly predicted 846 non–CHD cases (true negatives) and 898 CHD-positive cases (true positives). It also reported 154 false positives and 102 false negatives.

indicating excellent performance in detecting CHD. Specificity is:

$$\text{Specificity} = \frac{824}{824 + 176} \approx 82.4 \tag{39}$$

This shows that while Boosting is aggressive in minimizing false negatives (which is critical in medical diagnosis), it incurs a slightly higher false positive rate. The high sensitivity makes Boosting particularly effective in clinical scenarios requiring thorough screening.

**Bagging.** The sensitivity of the Bagging model is:

$$\text{Sensitivity} = \frac{882}{882 + 118} \approx 88.2 \tag{40}$$

and specificity is:

$$\text{Specificity} = \frac{830}{830 + 170} \approx 83.0 \tag{41}$$

Bagging shows a well–balanced trade–off between both classes. Although not as aggressive as Boosting, it is generally more robust to variance, offering reliable performance across different data samples, which is advantageous in real–world medical datasets.

**Stacking.** The sensitivity is:

$$\text{Sensitivity} = \frac{898}{898 + 102} \approx 89.8 \tag{42}$$

and the specificity is:

$$\text{Specificity} = \frac{846}{846 + 154} \approx 84.6 \tag{43}$$

These metrics indicate strong and stable performance, driven by the meta-learner's capacity to synthesize multiple decision boundaries. Stacking proves effective at improving the strengths of individual models, offering high sensitivity for clinical needs and strong specificity for reducing overdiagnosis.

### Confused matrix of enhanced/improved ensemble models

**Bayesian Model Averaging (BMA).** These results translate into strong diagnostic performance, with the BMA model demonstrating high sensitivity (recall) and strong specificity. Sensitivity is calculated as:

$$\text{Sensitivity} = \frac{867}{867 + 112} \approx 88.6\% \tag{44}$$

indicating the model's excellent ability to identify patients who truly have CHD. Specificity is:

$$\text{Specificity} = \frac{821}{821 + 158} \approx 83.8\% \tag{45}$$

reflecting its capacity to correctly dismiss non–CHD cases. However, the presence of 158 false positives might result in unnecessary further testing or anxiety for patients. On the other hand, the false negative count of 112 is comparatively low, which is crucial in clinical settings where missing true CHD cases can have severe consequences. Overall, the BMA ensemble model offers a reliable balance between minimizing both false negatives and false positives (Figs 14–18).

**Majority voting.** Compared to BMA, the Majority Voting model achieved a slightly better specificity:

$$\text{Specificity} = \frac{836}{836 + 143} \approx 85.4\% \tag{46}$$

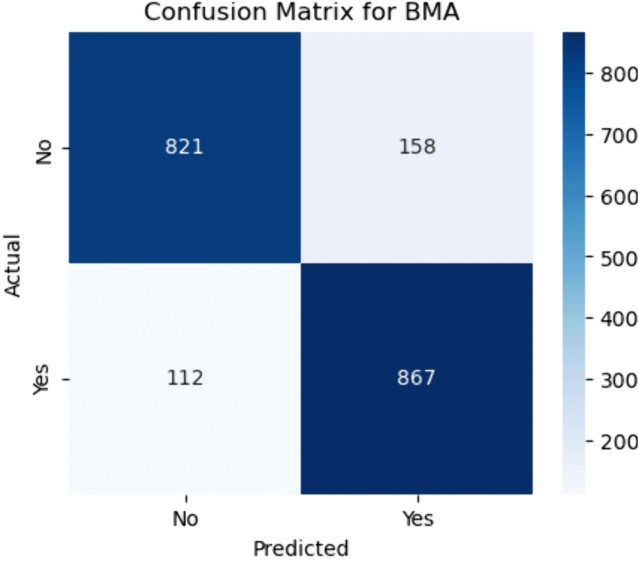

**Fig 14. Confusion matrix for Bayesian model averaging ensemble.** The figure shows the confusion matrix for the Bayesian Model Averaging (BMA) ensemble method, illustrating classification performance across all classes. Out of all actual negative cases (i.e., patients who did not have CHD), the model correctly predicted 821 as negative (true negatives) and misclassified 158 as positive (false positives). Conversely, among the actual positive cases (patients with CHD), it correctly identified 867 as positive (true positives), while misclassifying 112 as negative (false negatives).

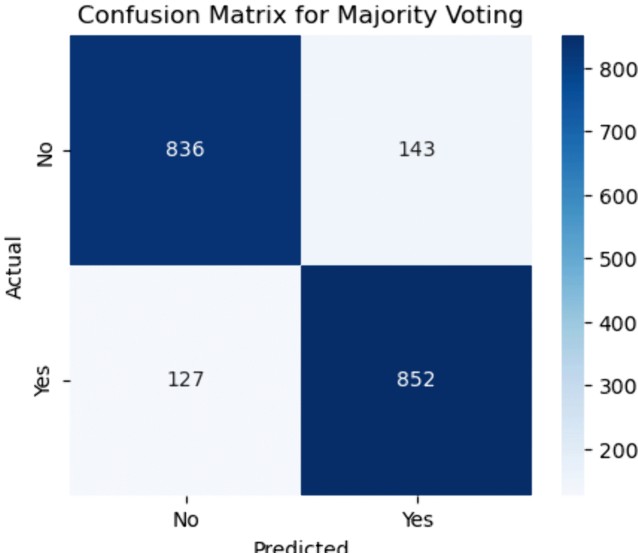

**Fig 15. Confusion matrix for majority voting ensemble.** The figure shows the confusion matrix for the Majority Voting ensemble method, depicting the distribution of correct and incorrect predictions. For actual non–CHD cases, this model correctly predicted 836 as negative (true negatives) and misclassified 143 as CHD (false positives). For actual CHD cases, 852 were accurately identified as positive (true positives), while 127 were misclassified as negative (false negatives).

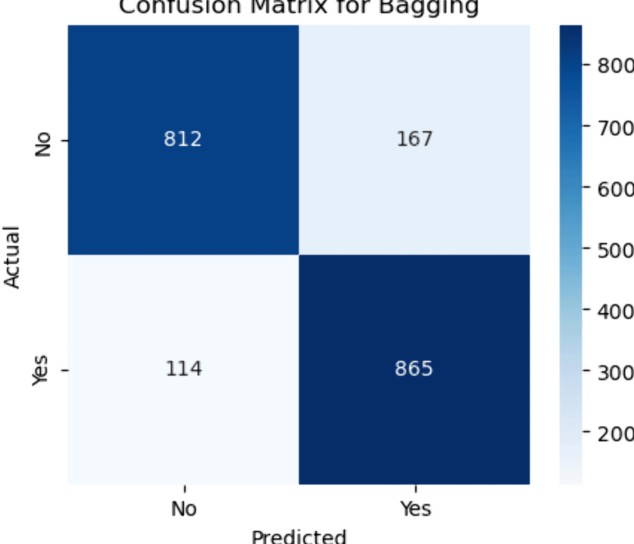

**Fig 16. Confusion matrix for bagging ensemble.** The figure shows the confusion matrix for the Bagging (Bootstrap Aggregation) ensemble model, highlighting the classification outcomes for all classes. Out of all actual negative cases (non–CHD), 812 were correctly classified as negative (true negatives), while 167 were misclassified as positive (false positives). Among actual positive CHD cases, 865 were correctly identified (true positives), and 114 were misclassified as negative (false negatives).

meaning it was marginally more effective at identifying non-CHD cases correctly. However, its sensitivity was lower:

$$\text{Sensitivity} = \frac{852}{852 + 127} \approx 87.0\% \tag{47}$$

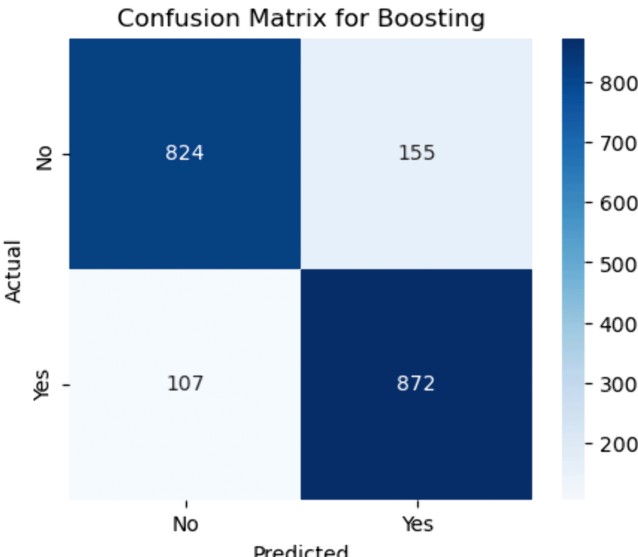

**Fig 17. Confusion matrix for boosting ensemble.** The figure shows the confusion matrix for the Boosting ensemble model, which delivers enhanced performance through sequential learning. The model correctly classified 824 of the non–CHD cases (true negatives) and incorrectly flagged 155 as positive (false positives). It also correctly detected 872 true CHD cases (true positives) and missed 107 positive cases (false negatives).

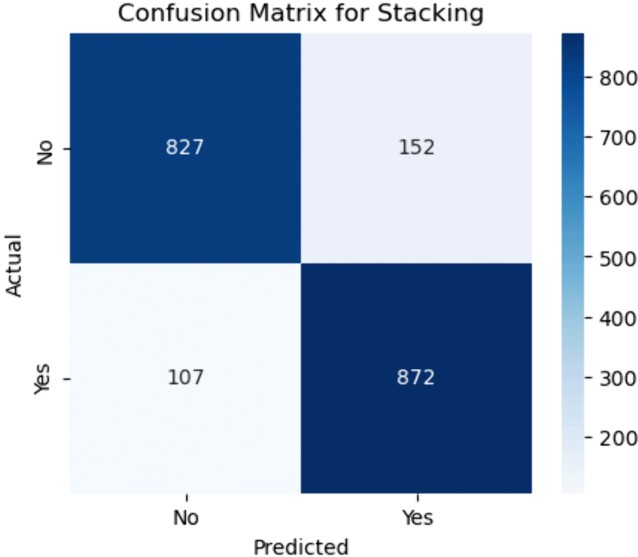

**Fig 18. Confusion matrix for stacking ensemble.** The figure shows the confusion matrix for the Stacking ensemble model, which emerged as the best-performing model in this study. From the matrix, the model correctly predicted 827 of the actual negative cases (true negatives) and misclassified 152 as positive (false positives). In the positive class (patients with CHD), the model correctly identified 872 as true positives and incorrectly labeled 107 cases as negatives (false negatives).

suggesting it missed slightly more true CHD cases than BMA. While this trade–off led to fewer false positives, it slightly increased the risk of failing to detect CHD in patients who actually have the condition. Clinically, this might be less favor-able in high-risk populations where identifying every CHD case is a priority, even at the expense of more false positives.

**Bagging.** The sensitivity (recall), calculated as:

$$\text{Sensitivity} = \frac{865}{865 + 114} \approx 88.3\% \tag{48}$$

indicates a strong ability to detect CHD cases. The specificity is given by:

$$\text{Specificity} = \frac{812}{812 + 167} \approx 82.9\% \tag{49}$$

This shows that the model performs well in ruling out non–CHD cases but does produce a fair number of false alarms. Bagging's effectiveness stems from its strategy of training multiple base learners on different bootstrap samples and averaging their predictions, which reduces variance and improves generalization.

However, the relatively higher false positive count (167) may lead to over-screening or unnecessary clinical follow–up for patients wrongly flagged as at-risk. Despite this, the model maintains a good balance between sensitivity and specificity, making it suitable for applications where slightly higher false positives are acceptable in exchange for fewer missed CHD diagnoses.

**Boosting.** The sensitivity of the Boosting model is:

$$\text{Sensitivity} = \frac{872}{872 + 107} \approx 89.1\% \tag{50}$$

slightly higher than Bagging, suggesting improved detection of CHD-positive patients. The specificity is:

$$\text{Specificity} = \frac{824}{824 + 155} \approx 84.2\% \tag{51}$$

also modestly better than Bagging's performance. These improved results confirm that Boosting, particularly models like Gradient Boosting or XGBoost, excels in reducing bias by focusing on the mistakes of previous learners during training iterations. The combination of higher sensitivity and specificity makes Boosting highly attractive for clinical screening scenarios where both early detection and minimizing overdiagnosis are critical. Furthermore, the relatively lower false negative count (107) compared to Bagging (114) further reinforces its utility in predictive healthcare applications.

**Stacking ensemble model.** These results reflect a high sensitivity (recall):

$$\text{Sensitivity} = \frac{872}{872 + 107} \approx 89.1\% \tag{52}$$

indicating that the stacking model is highly effective at correctly identifying CHD-positive individuals. The specificity, calculated as:

$$\text{Specificity} = \frac{827}{827 + 152} \approx 84.5\% \tag{53}$$

is also quite strong, suggesting good performance in recognizing true negative cases. These two metrics in tandem confirm that the model maintains a well-balanced classification boundary between the positive and negative classes—crucial in CHD screening, where both under-diagnosis and over-diagnosis can carry significant clinical risks. The relatively low number of false negatives (107) reinforces the model's suitability for early detection scenarios, while the false positive count (152) remains within an acceptable range for follow-up diagnostic assessment in clinical settings.

The stacking model benefits from its design, which integrates multiple base learners (e.g., logistic regression, random forest, SVM) into a meta-classifier that learns to correct individual model weaknesses. This fusion allows the model to

capture both linear and non-linear relationships in the data and leverage complementary decision boundaries from diverse algorithms. The high performance observed in this matrix confirms that stacking not only improves raw accuracy but also optimizes the sensitivity-specificity trade–off better than most standalone models or simpler ensembles like bagging or majority voting.

### Evaluating ML Models: Individual (Baseline) vs Improved Models for Predicting CHD

Individual (baseline) and enhanced ML models were evaluated for predicting CHD–this study utilized a refined and balanced dataset consisting of 9,789 observations obtained after applying SMOTE and stratified random sampling. The data was split using a 70% training and 30% testing ratio, and the models' predictive performances were assessed using five key classification metrics: Accuracy, Precision, Sensitivity (Recall), Specificity, and AUC-ROC as shown in Table 8. Both baseline ML models (Decision Tree, Random Forest, Gradient Boosting, and SVM) and improved ML models (ANRDT, HIRF, PGBM, and ESVM) were evaluated to determine how algorithmic enhancements influence predictive accuracy and robustness in the context of CHD.

### Baseline machine learning models

Gradient Boosting (GBM) and Random Forest (RF) emerged as top performers among the baseline models. GBM achieved the highest sensitivity (0.908), indicating a superior ability to identify true positives (patients with CHD), while RF demonstrated a slightly higher AUC-ROC (0.937), denoting excellent discriminative capability. Both models outperformed Decision Tree (DT) and Support Vector Machine (SVM) across nearly all metrics. The Decision Tree, though interpretable, yielded relatively modest performance with an accuracy of 0.807 and a specificity of 0.802, reflecting a limited generalization capacity. While achieving an accuracy of 0.830 and a high AUC-ROC of 0.906, SVM fell short compared to the ensemble approaches in capturing complex patterns in the data.

### Enhanced machine learning models

The enhanced models showed how specific changes proved effective in overcoming the shortcomings of their original versions. Interestingly, the Pruned Gradient Boosting Machine (PGBM) was found to be equivalent to the baseline GBM by accuracy (0.866). It outperformed precision (0.849) and specificity (0.842), showing more balanced performance, fewer false positives, and better generalization. The Hybrid Imbalanced Random Forest (HIRF), based on SMOTE and SHAP values, had a marginally lower overall precision (0.856) but had a better balance of sensitivity (0.884) and specificity (0.829), indicating its effectiveness in class imbalance.

**Table 8**. Comparison of baseline and enhanced machine learning models on classification metrics.

| Model | Accuracy | Precision | Sensitivity | Specificity | AUC-ROC |
|---|---|---|---|---|---|
| *Baseline Models* | | | | | |
| Decision Tree | .807 | .804 | .811 | .802 | .807 |
| Random Forest | .863 | .845 | .888 | .837 | .937 |
| Gradient Boosting | .866 | .838 | .908 | .824 | .941 |
| Support Vector Machine | .830 | .817 | .849 | .810 | .906 |
| *Enhanced Models* | | | | | |
| ANRDT (Enhanced DT) | .862 | .848 | .883 | .842 | .933 |
| HIRF (Enhanced RF) | .856 | .838 | .884 | .829 | .928 |
| PGBM (Enhanced GBM) | .866 | .849 | .891 | .842 | .937 |
| ESVM (Enhanced SVM) | .816 | .790 | .861 | .771 | .902 |

Table notes: $N = 9,789$. Performance metrics are calculated on a 70% training and 30% testing data split. Sensitivity and specificity refer to the model's ability to correctly identify positive and negative cases, respectively.

Likewise, the Adaptive Noise-Resistant Decision Tree (ANRDT) has shown significant improvement compared to the baseline decision trees (DT), with improvements seen in all the metrics, namely, accuracy (0.862), sensitivity (0.883), and AUC-ROC (0.933). These improvements bear out the hypothesis that probabilistic splits and regularization lowered over-fitting and made them more robust to noisy inputs. Although the Enhanced SVM (ESVM) showed an increase in sensitivity (0.861) and the interpretability of the SHAP integration, it did not perform better than the baseline SVM across all measures. This finding can be attributed to the sensitivity of ESVM to the choice of a kernel and lower specificity (0.771), signalling the difficulty of achieving trade-offs in SVM tuning.

In general, the improved models performed as well as or better than the base ones in all crucial measures, particularly in sensitivity and AUC-ROC, which are essential in the detection of CHD cases. Pruning, reweighting and noise resistance are all ensemble-based improvement methods that offer indicated generalizability, robustness and interpretability advantages, which are key factors in clinical decision-making.

### Scalability analysis of baseline models

In addition to predictive performance, we evaluated scalability metrics of the baseline models to assess deployment feasibility in low-resource settings. Table 9 summarizes training time, inference latency, model size, and memory usage across Decision Tree, Random Forest, Gradient Boosting, and SVM classifiers.

The results reveal that the **Decision Tree** achieved the fastest training time (~0.09 s) and negligible inference latency, with a compact model size of 176 KB and minimal memory usage, making it highly lightweight. In contrast, **Random Forest** and **Gradient Boosting** required longer training times (~1.7–2.0 s), though both maintained near-zero inference latency. However, Random Forest exhibited a considerably larger storage footprint (~19 MB) and higher memory consumption compared to Gradient Boosting (~192 KB, 0.76 MB).

The **Support Vector Machine (SVM)** demonstrated the least scalable profile, requiring over 35 seconds for training, the highest inference latency (0.0018 s per sample), and substantial memory usage (~39 MB), despite a moderate model size of 1 MB. These findings suggest that, while SVM can deliver strong predictive performance, it may be less suitable for real-time or resource-constrained deployments.

Overall, the analysis highlights a trade-off between accuracy and scalability. Ensemble methods (Random Forest, Gradient Boosting) achieved superior predictive performance but incurred higher storage and memory costs. Simpler models such as Decision Trees offered lower accuracy but excelled in speed and efficiency. This underscores the importance of balancing predictive accuracy with computational feasibility when considering model deployment in clinical environments.

### ROC–AUC curve for baseline and improved ML models

A cross–comparison between the two ROC plots reveals that the enhanced models exhibit more tightly grouped and consistently high-performing ROC curves, reflecting their greater robustness and stability across varying thresholds. The improvements in AUC, especially in ANRDT and HIRF, underscore the success of tailored model enhancements in addressing key limitations such as overfitting, class imbalance, and lack of interpretability. These results reinforce the value of hybridization and model optimization strategies in developing machine learning frameworks suited for CHD prediction (Figs 19 and 20).

**Table 9**. Scalability metrics for baseline machine learning models.

| Model | Train Time (s) | Inference Latency (s/sample) | Model Size (KB) | Train Memory (MB) | Inference Memory (MB) |
|---|---|---|---|---|---|
| Decision Tree | 0.085 | 0.0000 | 176 | 1.98 | 0.00 |
| Random Forest | 1.738 | 0.0000 | 19,737 | 4.36 | 0.38 |
| Gradient Boosting | 2.002 | 0.0000 | 193 | 0.76 | 0.16 |
| Support Vector Machine | 35.264 | 0.0018 | 1,035 | 39.33 | 0.00 |

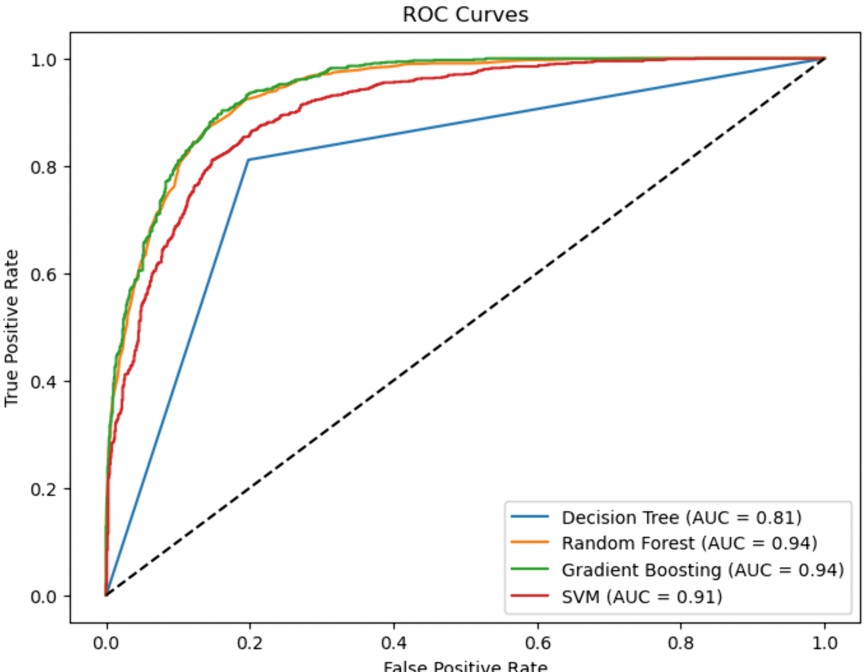

**Fig 19**. **ROC–AUC curve for individual (baseline) ML models.** The figure shows the ROC curves and corresponding Area Under the Curve (AUC) values for four baseline machine learning models—Decision Tree, Random Forest, Gradient Boosting, and Support Vector Machine (SVM)—demonstrating their discriminatory ability in predicting CHD. Of these, the Gradient Boosting and Random Forest models have the highest AUC scores of 0.94, which means that they have a very good chance of separating the CHD–positive and CHD–negative cases. SVM also performs well with an AUC of 0.91, though it lags slightly behind the ensemble models. The Decision Tree model, by contrast, demonstrates the weakest performance, with an AUC of 0.81 and a visibly lower curve, indicating reduced sensitivity and specificity. This underperformance may be attributed to its higher variance and sensitivity to noise in the data.

### Precision–recall curves

By contrast, the Random Forest and Gradient Boosting models perform better, with the highest AP values of 0.93 and 0.94, respectively. Their PR curves are comparatively constant below and near the peak for every level of recall and show a perfect balance of precision and recall. This implies that the two models perform well in the detection of real CHD cases and false positive reduction. The Support Vector Machine (SVM) is also doing a good job as well with an AP of 0.90. Although it is a little below the ensemble models, the SVM has high precision in the majority of the recall thresholds, indicating good and sturdy performance in classification (Figs 21 and 22).

PGBM maintains the best performance level with an AP of 0.93, equal to the baseline model Gradient Boosting. The pruning methods and powerful loss functions employed by PGBM enable the model to retain precision and enhance generalizability and interpretability. ESVM is not significantly different with the AP of 0.90 compared to the baseline SVM. The slight dip in average precision may be attributed to the trade–offs introduced by interpretability enhancements and automated tuning, yet the curve still indicates stable and reliable classification performance.

### Calibration curves for baseline and enhanced ML models

In Fig 23, the calibration performance of the baseline models–Decision Tree (DT), Random Forest (RF), Gradient Boosting (GBM), and Support Vector Machine (SVM)–is illustrated. The Decision Tree exhibits notable miscalibration, with its curve consistently deviating above the diagonal line. This indicates overconfidence in its predictions, where predicted probabilities are higher than actual observed outcomes. This is consistent with the model's relatively lower performance

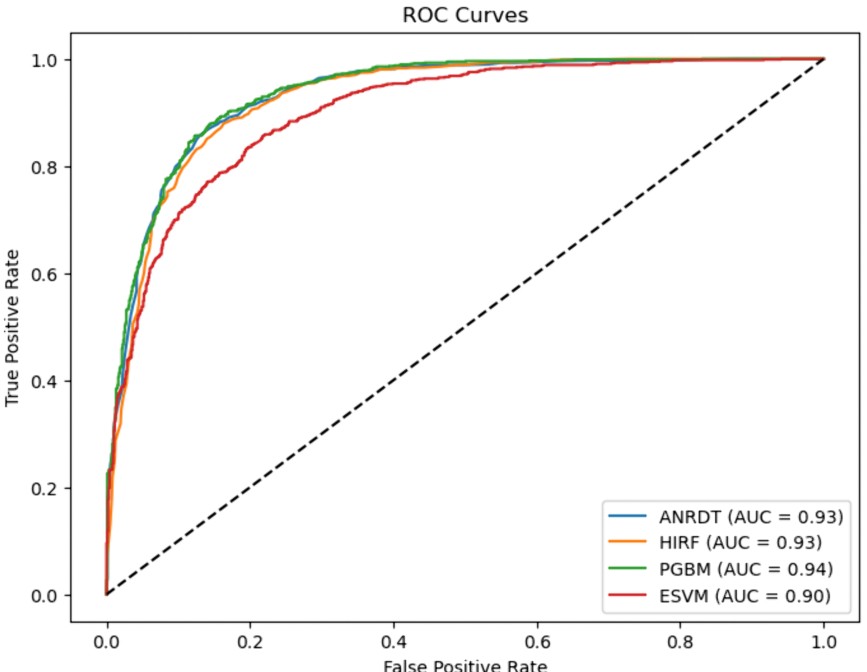

**Fig 20**. **ROC–AUC curve for enhanced ML models.** The figure shows the ROC curves and corresponding AUC values for the enhanced versions of the baseline models: ANRDT (Adaptive Noise-Resistant Decision Tree), HIRF (Hybrid Imbalanced Random Forest), PGBM (Pruned Gradient Boosting Machine), and ESVM (Enhanced Support Vector Machine), highlighting improvements in predictive performance. The enhancements to the Decision Tree model (ANRDT) led to a remarkable performance boost, with the AUC increasing from 0.81 to 0.93. This suggests that soft splits, pruning, and regularization significantly enhance the model's robustness and generalization capabilities. Similarly, HIRF achieves an AUC of 0.93, nearly matching the baseline Random Forest, while addressing class imbalance through SMOTE and improving interpretability with SHAP value integration. PGBM retains the high AUC of 0.94 seen in the baseline Gradient Boosting model, indicating that the pruning and outlier resistance techniques did not compromise predictive performance but maintained it while improving model reliability. ESVM, while slightly lower than the baseline SVM with an AUC of 0.90, still demonstrates competitive performance, suggesting that the trade-off for improved interpretability and automated hyperparameter tuning may have come at a minimal cost to classification power.

across accuracy and AUC–ROC metrics. In contrast, both Random Forest and Gradient Boosting maintain calibration curves closer to the diagonal, particularly at mid to high probability ranges, suggesting reliable probability estimates. The SVM model demonstrates moderate calibration, with a slight tendency to underestimate probabilities, particularly in the mid-range. This conservative calibration is characteristic of SVMs and contributes to their generally stable but sometimes less sensitive classification behavior.

Fig 24 shows the calibration curves of the enhanced models Adaptive Noise-Resistant Decision Tree (ANRDT), Hybrid Imbalanced Random Forest (HIRF), Pruned Gradient Boosting Machine (PGBM), and Enhanced Support Vector Machine (ESVM). These models are better calibrated than their baseline counterparts. ANRDT shows substantial improvement upon the contrast of baseline DT, with its calibration curve being a far better fit with the ideal diagonal. This symbolizes the optimistic effect of reinforcements like probabilistic splitting, noise filtering and regularization in alleviating overfitting and enhancing the reliability of predictions. HIRF and PGBM are also well-calibrated and highly coherent with the diagonal over the range of probabilities. These enhancements align with their good classification responsiveness and add that the algorithmic improvement is not only accuracy improvements but also confidence estimation. ESVM calibration profile is close to the baseline SVM with marginal underestimation of probabilities despite its overall reliable performance.

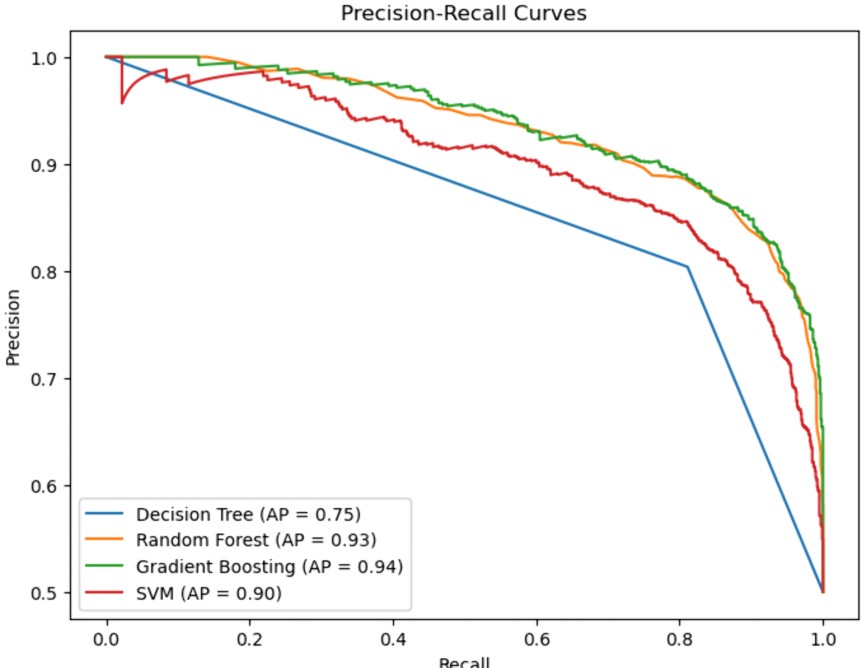

**Fig 21. Precision–recall curve for individual (baseline) ML models.** The figure shows the precision-recall (PR) curves and corresponding Average Precision (AP) values for four baseline machine learning models—Decision Tree, Random Forest, Gradient Boosting, and Support Vector Machine (SVM)—providing a detailed assessment of model performance in predicting CHD, particularly for imbalanced datasets. The decision Tree model has the lowest performance among the four, with an AP of 0.75. When the recall is higher, its PR curve collapses, and hence, the model is unable to retain its precision when classifying more positive instances. This represents a blatant indication that the Decision Tree produces a relatively large number of false positives, a factor which renders it unsuitable for clinical practice.

## Learning curves for baseline VS. enhanced ML models

### Baseline ML models

### Enhanced ML models

Learning curves are valuable diagnostic tools for understanding a model's performance as the size of the training dataset increases. They display two key metrics: the training score (performance on the training set) and the cross-validation (CV) score (performance on unseen data), both plotted against training set size. A large, persistent gap between these curves indicates overfitting, while convergence suggests effective generalization (Figs 25–32).

## Scalability analysis of enhanced models

Beyond predictive performance, we formally assessed the scalability of the enhanced machine learning models. Table 10 summarizes the results for training time, inference latency, model size, and memory usage. This provides a comprehensive view of computational feasibility, particularly for deployment in resource-limited healthcare environments.

The **ANRDT** achieved a favorable balance, with modest training time ($\sim$0.93 s), negligible inference latency, and a relatively compact model size ($\sim$19 MB). Its memory footprint was also moderate, requiring approximately 6 MB during training and less than 0.3 MB for inference. Similarly, **PGBM** and **Boosting** were computationally efficient, completing training in under 2 seconds, with very small model sizes ($\sim$192 KB) and minimal memory requirements, making boosting-based models particularly attractive for deployment on constrained devices.

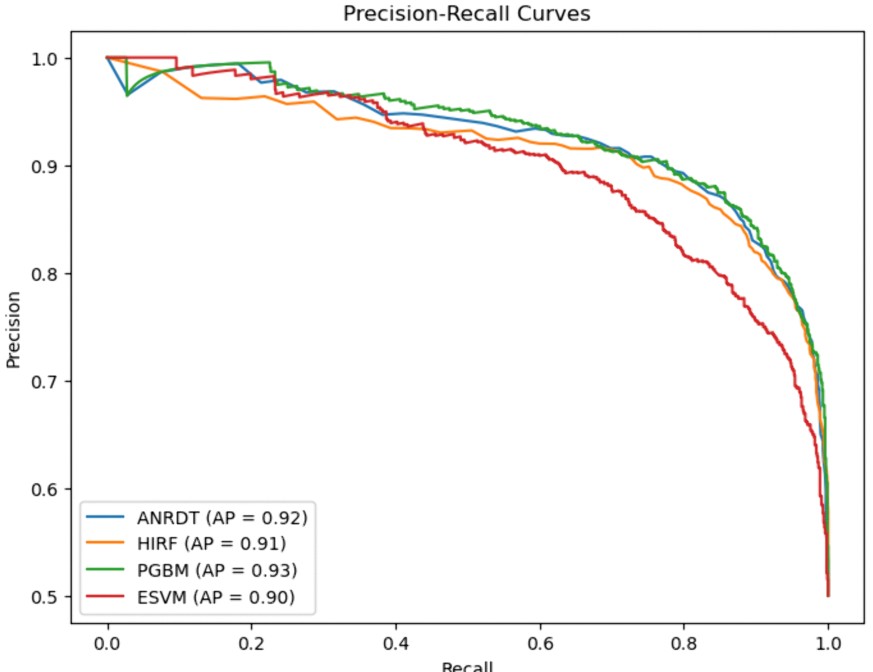

**Fig 22**. **Precision–recall curve for enhanced ML models.** The figure shows the precision-recall (PR) curves and corresponding Average Precision (AP) values for the enhanced versions of the baseline models: ANRDT (Adaptive Noise-Resistant Decision Tree), HIRF (Hybrid Imbalanced Random Forest), PGBM (Pruned Gradient Boosting Machine), and ESVM (Enhanced Support Vector Machine), highlighting improvements in predictive accuracy for imbalanced data. ANRDT excels with a significant change, increasing the AP to 0.92 as compared to the baseline model of 0.75. The positive change highlights the usefulness of noise filtering, pruning, and regularization enhancements to enhance the precision of the Decision Tree and decrease false positives. HIRF also shows a good performance with an AP of 0.91. Though with a slighter decrease than its baseline analogue, the improved model is stable and reliable, particularly when it comes to working with imbalanced data and synthetic oversampling alongside weighted learning.

**HIRF** and **Bagging** required longer training times (2.8 s and 4.5 s, respectively) and moderate model sizes (~11 MB), but remained scalable with low inference memory. In contrast, **ESVM** exhibited the slowest profile, requiring over 20 seconds of training, higher inference latency (~0.001 s/sample), and the largest training memory usage (29 MB), which limits its scalability.

The most computationally demanding approach was **Stacking**, which required over 140 seconds of training and resulted in a large model size (~65 MB), along with significant training memory usage (~17 MB). Although stacking achieved the highest predictive performance, its heavy computational and memory requirements may constrain feasibility in real-time or resource-limited settings.

Finally, the ensemble aggregation approaches, **BMA** and **Majority Voting**, incurred negligible training overhead but required relatively high inference latency (~0.0010–0.0012 s/sample). Their model sizes were relatively large (up to ~32 MB) and inference memory usage was non-trivial (1.1 MB for BMA), underscoring that ensemble simplicity does not always translate to computational efficiency.

Overall, these results show that while enhanced models such as stacking and bagging offer incremental performance gains, boosting-based approaches provide the most favorable trade-off between predictive accuracy, model compactness, and memory efficiency. The inclusion of RAM usage metrics further demonstrates their practicality for deployment in low-resource healthcare environments.

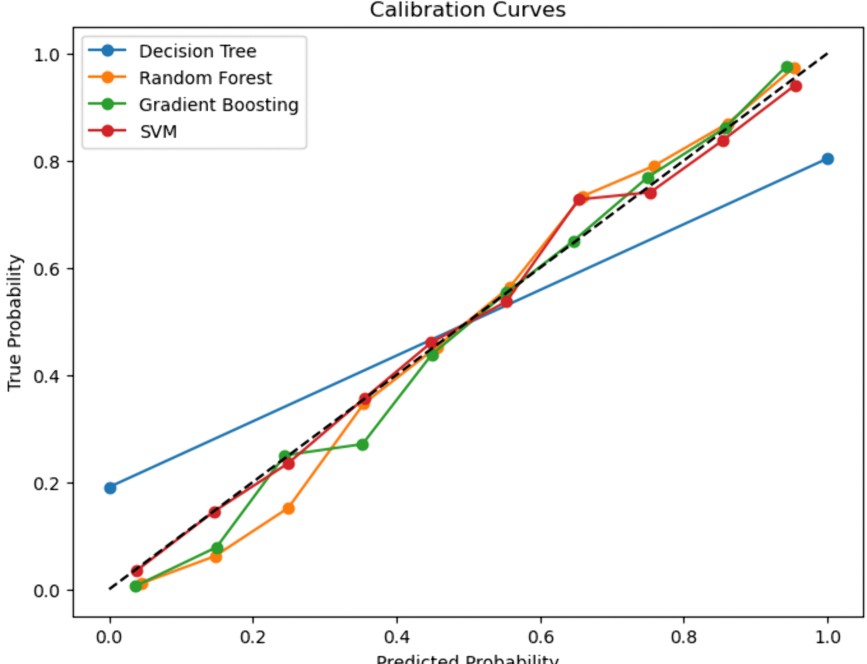

**Fig 23**. **Calibration curve for baseline ML models.** The figure shows the calibration curves for baseline machine learning models, illustrating how closely the predicted probabilities of CHD align with the actual observed outcomes, thereby assessing model reliability.

## Comparison of learning curves: Baseline vs. enhanced models

The comparison of the learning curves of the baseline and the improved machine learning models gives a critical insight into the generalization behavior of the models and the ability to overcome overfitting. In the baseline models, the Decision Tree (DT) performs excellently on the training set with all sample sizes, implying that it is overfitting the data. Nevertheless, its cross-validation (CV) score is much lower, and it has small incremental training progress. The tendency of the training performance to remain high but the validation performance to remain lower is a clear sign of overfitting, and it is indicative that the model fits the noise and certain patterns specific to the training data and fails to generalize to the validation set. The Random Forest (RF) model also does not overfit quite as much as the Decision Tree (DT), with the gap between the training and the validation scores being much narrower. This can be explained by its ensemble architecture that integrates variance-reduction schemes, which are bootstrapping and random feature selection. However, the plateau in the validation score implies that the potential of RF will not be high until it is tuned or artifacts are added to the data.

Gradient Boosting Machine (GBM) has a better learning dynamic. With an increase in the size of the training set, the training accuracy of GBM is minimally reduced, whereas the CV score is consistently increased. The trend indicates the ability of the model to learn sequentially by drawing on previous errors and thus generalizing better than DT and RF. Nevertheless, the final difference between the training and validation scores suggests that GBM will be prone to overfitting unless adequately regularized. The Support Vector Machine (SVM) indicates the most balanced performance among the models that form the baseline analysis. Its learning curves, training, and validation have relatively low starting but maintain the same monotonic growth pattern. The decrease in the distance between them indicates a high level of generalization and indicates that SVM heuristically manages the bias-variance problem well, at least when used with a proper kernel and regularisation strategy.

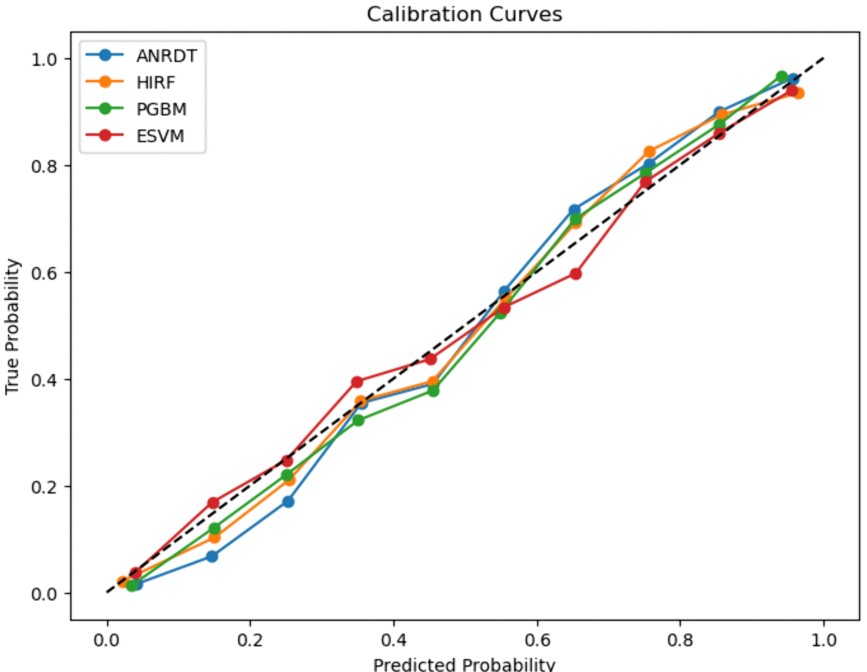

**Fig 24**. **Calibration curve for enhanced ML models.** The figure shows the calibration curves for the enhanced versions of the baseline models, highlighting improvements in the alignment between predicted probabilities and actual CHD outcomes, indicating more reliable probability estimates. Ideally, a model's calibration curve aligns with the diagonal 45–degree line, indicating perfect correspondence between predicted and observed probabilities.

On the contrary, the improved models have better generalization behavior and improved learning when additional training data is provided. The Adaptive Noise-Resistant Decision Tree (ANRDT) reflects the overfitting pattern of the baseline DT. Accordingly, the Adaptive Noise-Resistant Decision Tree also reached a perfect training score, with cross-validation (CV) performance being relatively flat. Although designed to be robust against noise, ANRDT generalizes poorly, suggesting that it might be necessary to augment it with some further mechanism, e.g., pruning or regularisation, before it can convert its memorization ability into prediction accuracy. Nevertheless, the Hybrid Imbalanced Random Forest (HIRF) demonstrates a more promising dynamic. Although it continues to perform well in terms of training accuracy, its validation performance improves more consistently as compared to ANRDT. This implies that its improvements in dealing with imbalanced data, possibly as a result of weighting or superior sampling methods, will be a contributor to improved generalization.

An exciting improvement is noted in the Pruned Gradient Boosting Machine (PGBM), displaying a favorable balance of bias-variance tradeoff. As the training set size rises, the accuracy of the model in training gradually drops, but the cross-validation score constantly improves and then reaches a plateau slightly below the training curve. This overlap shows efficient pruning and regularization that enables PGBM to attain high accuracy without overfitting. The comparison of PGBM and its baseline version shows a more stable and generalizable result. Looking at it the same way, Enhanced Support Vector Machine (ESVM) also has a good generalization ability. After starting both training and validation scores at low levels, the difference in the two scores is seen to decrease with an increase in data, indicating a close convergence. This trend suggests that ESVM is capable of rectifying the underlying configuration of the data rather than being subjected to noise, probably because of better regularization and a better selection of kernels.

Generally, the improved models are shown to enhance the learning efficiency and the generalization ability of the model over the baseline models. Although techniques to improve ensembles, such as HIRF, have some effect in

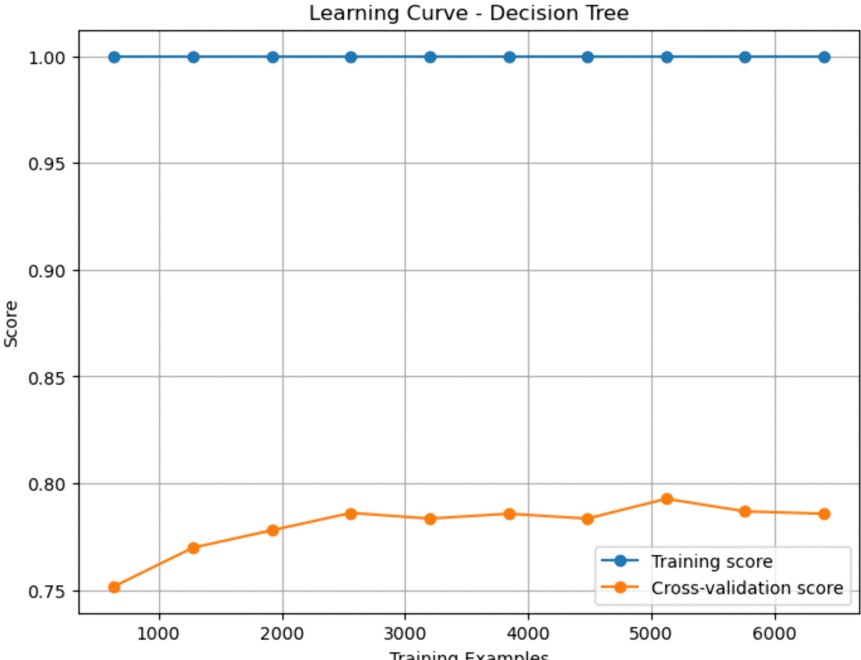

**Fig 25. Learning curve for decision tree.** The figure shows the learning curve for the Decision Tree (DT) model. The training score remains at a perfect 1.00 across all training set sizes, indicating that the model fully memorizes the training data, which may suggest overfitting. In contrast, the cross-validation (CV) score starts relatively low, at around 0.75, and increases modestly to approximately 0.78. Despite this slight improvement, a substantial gap persists between the training and validation scores, clearly demonstrating overfitting. The DT model captures intricate patterns in the training data but fails to generalize to unseen data. This behavior is typical of unpruned or overly complex decision trees, which are prone to learning noise, resulting in high variance. The limited gain in validation performance with increasing data suggests that additional training samples alone are insufficient to resolve overfitting. Instead, techniques such as pruning, regularization, or switching to ensemble approaches may be necessary to enhance generalization.

discouraging overfitting, much greater improvements can be achieved with more focused modifications, such as pruning, boosting optimization, and regularisation, which are observed in models that place a stronger emphasis on model customization, such as PGBM and ESVM. Such findings highlight the importance of complexity control and robustness strategies in model improvement as a means of constructing solid and scalable predictive systems in medicine, fields like predicting coronary heart disease risk.

## Ensemble learning models

The performance comparison between baseline and enhanced ensemble learning models reveals important patterns in how each approach contributes to coronary heart disease (CHD) prediction. All ensemble methods exhibit high predictive capacity, with accuracies generally above 0.85. Among the baseline models, stacking achieves the highest overall accuracy at 0.872, followed closely by boosting at 0.866. Stacking also demonstrates a strong balance across precision (0.854), sensitivity (0.898), and specificity (0.846), indicating its robustness in classifying both positive and negative CHD cases. Boosting, although slightly lower in precision and specificity, yields the highest sensitivity (0.908), highlighting its strength in correctly identifying positive CHD cases, albeit at a slight cost to false-positive rates.

Majority voting also performs competitively, with an accuracy of 0.864, as well as well-balanced precision (0.857) and specificity (0.854). Its relatively simple decision mechanism makes it effective for stabilizing predictions across diverse base learners. Bayesian Model Averaging (BMA), though theoretically powerful due to its probabilistic weighting of model

**Fig 26**. **Learning curve for random forest.** The figure shows the learning curve for the Random Forest (RF) model. Similar to DT, the training score remains close to 1.00, reflecting the model's capacity to fit the training set well. However, the CV score starts higher than in DT, around 0.84, and improves gradually to about 0.86. While there is still a gap between training and validation scores, it is narrower compared to DT, indicating reduced overfitting. This improvement is attributed to the ensemble nature of RF, which leverages bagging and random feature selection to reduce variance. Although some overfitting remains, the model generalizes better than DT, making RF a more reliable baseline model. The plateauing of the CV score suggests that further enhancements may require hyperparameter tuning (e.g., number of estimators, tree depth) or additional feature engineering.

outputs, delivers slightly lower performance (accuracy of 0.858), possibly due to challenges in estimating accurate posterior probabilities with limited data. Bagging, on the other hand, while slightly underperforming in precision and specificity compared to other methods, offers the second-highest sensitivity (0.882), making it valuable when minimizing false negatives is critical.

The enhanced ensemble models, which are built by combining optimized versions of individual base learners, show marginal but meaningful improvements or stability in performance. The enhanced stacking model maintains strong performance across all metrics, achieving an accuracy of 0.868, a precision of 0.852, and balanced recall and specificity of 0.891 and 0.845, respectively. This suggests that integrating improved base learners helps retain stacking's advantage in generalization. Enhanced boosting also sustains its high sensitivity (0.891), indicating its continued effectiveness in recognizing CHD cases after enhancement, while its specificity (0.842) improves compared to the baseline. Similarly, enhanced majority voting maintains its original accuracy (0.862) and shows a slight improvement in precision, reinforcing the utility of simple ensemble strategies when paired with stronger individual classifiers.

Interestingly, enhanced BMA sees a slight uptick in accuracy (from 0.858 to 0.862) and sensitivity (from 0.876 to 0.886), suggesting that when better-calibrated models are used as inputs, BMA's probabilistic framework becomes more effective. Enhanced bagging, however, maintains a similar performance profile to its baseline counterpart, with nearly identical metrics. This implies that the improvements in individual learners may not substantially alter bagging's inherent variance-reduction mechanism unless its architecture is further modified (e.g., through feature selection or deeper trees).

Table 11 presents the detailed classification metrics for both baseline and enhanced ensemble models.

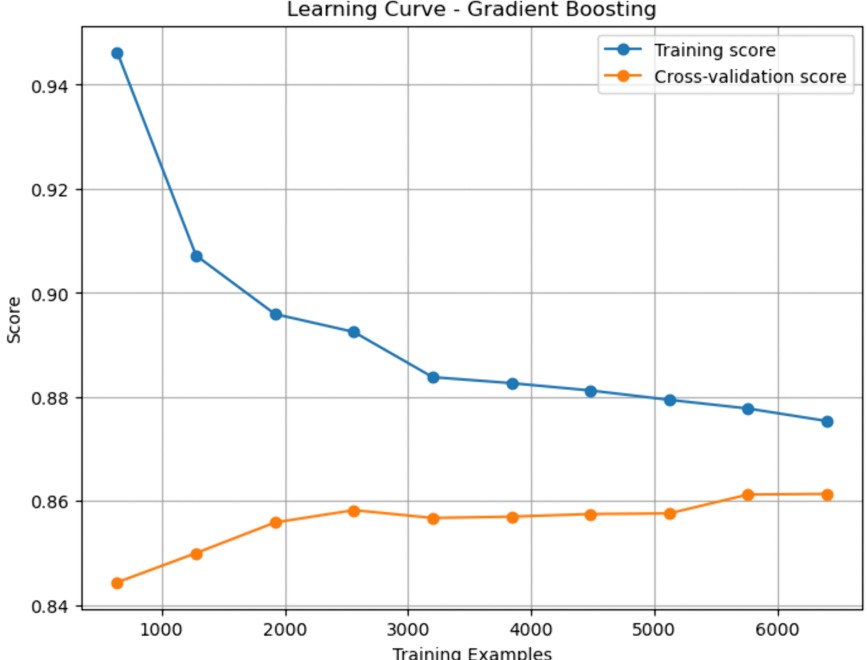

**Fig 27**. **Learning curve for gradient boosting machine.** The figure presents the learning curve for the Gradient Boosting Machine (GBM) model. The training score starts high at around 0.945 and gradually declines to just below 0.88 as more training data is added. This downward trend indicates that the model becomes less dependent on specific patterns in the training data and starts learning more generalizable relationships. In parallel, the CV score begins at approximately 0.845 and steadily increases, plateauing near 0.861. The moderate and decreasing gap between training and validation scores suggests that GBM manages overfitting more effectively than DT or RF. Unlike RF, GBM builds trees sequentially, with each new learner correcting the errors of the previous ones. This sequential learning process enhances accuracy but can also introduce sensitivity to noise. The stable CV score after a certain data threshold implies that the model has captured most of the useful patterns in the data. Further improvement would likely depend on regularization strategies such as shrinkage, subsampling, or adjusting the number of boosting iterations and learning rate.

## Comparative evaluation of machine learning and ensemble models for CHD prediction

The comparative evaluation of ML models, and ensemble learning models for predicting CHD reveals critical insights into the strengths and trade-offs of each modelling approach.

### Individual (baseline) machine learning models

Among the baseline machine learning models, Random Forest, Gradient Boosting, and Support Vector Machine (SVM) achieved competitive results. Gradient Boosting reached an accuracy of 86.6%, with high sensitivity (90.8%) but slightly lower precision (83.8%) and specificity (82.4%). Random Forest followed closely with an accuracy of 86.3% and more balanced metrics. SVM, while somewhat less accurate (83.0%), offered reasonably high sensitivity (84.9%) and precision (81.7%), demonstrating its utility in balanced classification tasks.

Enhanced versions of these models showed moderate but meaningful improvements. The Pruned Gradient Boosting Machine (PGBM), for example, preserved the original model's high performance with improved precision (84.9%) and specificity (84.2%). The Adaptive Noise-Resistant Decision Tree (ANRDT) significantly enhanced baseline decision trees, achieving an accuracy of 86.2%–approaching ensemble-level performance. These findings highlight the potential of model-specific enhancements to close the performance gap with more complex ensembles while retaining interpretability and efficiency.

**Fig 28**. **Learning curve for support vector machine.** The figure illustrates the learning curve for the Support Vector Machine (SVM) model. Both training and CV scores begin relatively low—around 0.66 and 0.63, respectively—but exhibit a consistent upward trend with increasing training size. The training score gradually rises to approximately 0.825, while the CV score closely follows, ending around 0.82. The consistently narrow gap between the two curves indicates low variance and strong generalization. This is characteristic of SVMs, which, when properly regularized and used with suitable kernels, effectively manage the bias-variance tradeoff. Unlike tree-based models, which tend to overfit, SVMs aim to maximize the margin between classes, which helps prevent overfitting, particularly in high-dimensional or noisy datasets. The steady improvement in CV performance suggests that the SVM model benefits significantly from additional training data and has not yet exhausted its learning capacity.

## Ensemble learning models

The obtained findings indicate that ensemble learning models outperform classical statistical models and even individual machine learning algorithms in predicting CHD. These models take advantage of the fact that combining the various learners- each with their predictive benefits- will result in a greater and superior overall model. The best performing model among all the models was stacking. The baseline stacking model achieved 87.2% accuracy, 85.4% precision, 89.8% sensitivity, and 84.6% specificity which depict a high overall balance and there was minimal trade-off between false positive and false negative. These tests show that not only does stacking work accurately but also has proven to be clinically reliable particularly in medical applications where false and true diagnosis is vital. This performance benefit was preserved in the improved stacking model, where an accuracy of 86.8% and an adjusted sensitivity of 89.1% were observed, demonstrating the robustness of the approach even with tuning and feature optimization.

The other important ensemble technique, boosting, also performed wonderfully on its default and improved versions. Among the tested models, Gradient Boosting Machines (GBM) in the ensemble structure gave the best sensitivity of 90.8%. The resulting high sensitivity is critical in CHD–screening and triage where any failure to identify high-risk individuals would lead to catastrophic consequences- an advantage of boosting is that it progressively improves the errors of the underlying model, allowing the ensemble to concentrate on those that are harder to classify- an advantage of imbalanced data, such as BRFSS. The accuracy remained the same (86.6%), and precision and specificity were modestly improved, showing that boosting algorithms can be improved by preprocessing measures such as pruning and class weighting.

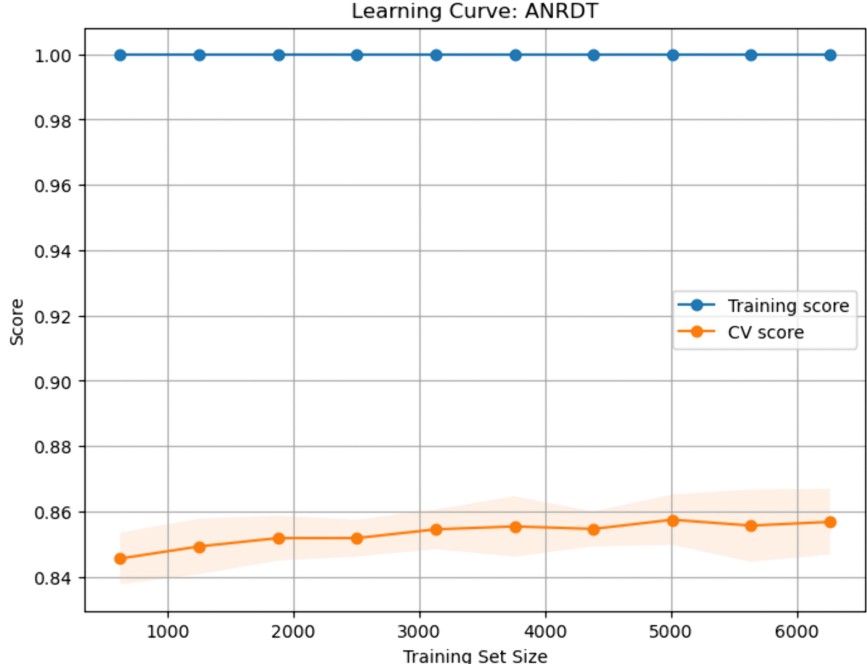

**Fig 29**. **Learning curve for ANRDT.** The figure shows the learning curve for the Adaptive Noise-Resistant Decision Tree (ANRDT) model. The training score remains consistently at 1.00 across all training set sizes, indicating that the model perfectly fits the training data. This reflects the model's ability to fully memorize the training data. However, the CV score starts at approximately 0.845 and rises only slightly to around 0.857. The wide and unchanging gap between the training and CV scores clearly indicates a high degree of overfitting, suggesting that ANRDT struggles to generalize to new data. Although there is a minor improvement in CV performance with more data, it is insufficient to offset the overfitting. To improve generalization, strategies such as regularization or pruning would be necessary.

Boosting can be an important choice as a standalone tool or in larger predictive pipelines, as it is highly reliable and also highly scalable.

Other ensemble methods, including Bayesian Model Averaging (BMA), Majority Voting, and Bagging, performed reliably and steadily as well. BMA obtained an accuracy of 85.8% for the baseline model and 86.2% for the best model, having similar precisions and specificities, demonstrating stability. Majority Voting also has a comparable precision of over 85% and sensitivity near 87% as well, thus being a viable alternative when simplicity and explainability are required. Bagging models are less precise and specific than other ensembles, but they have high sensitivity with both the baseline (88.2%) and the enhanced form (88.5%), so they are appropriate when it is most important to identify real positive cases. These results demonstrate that even primitive ensembles can dramatically outperform individual learners through their ability to decrease variance and balance a model bias, which is of special importance in clinical data where the variables may be noisy or correlating.

The high performance of the ensemble models can be attributed to a few important benefits. One is that they minimize the problem of overfitting by averaging the decisions of many different learners, which regularizes extreme predictions and reduces outlier impacts. Second, they increase generalization on unseen data, which can be proven by their stability in cross-validation folds in this evaluation. Third, ensemble methods are flexible and hence may enable application of interpretable (e.g., logistic regression), high-scoring black-box (e.g., GBM, SVM) learners, which is beneficial in a hybrid clinical setting. Improvement of the base learners utilized in this paper, such as Adaptive Noise–Resistant Decision Tree (ANRDT) and Pruned Gradient Boosting Machine (PGBM) is another benefit in regard to improvement of ensemble reliability due to insensitivity to noise and efficient handling of class imbalances. The stability of the ensemble models

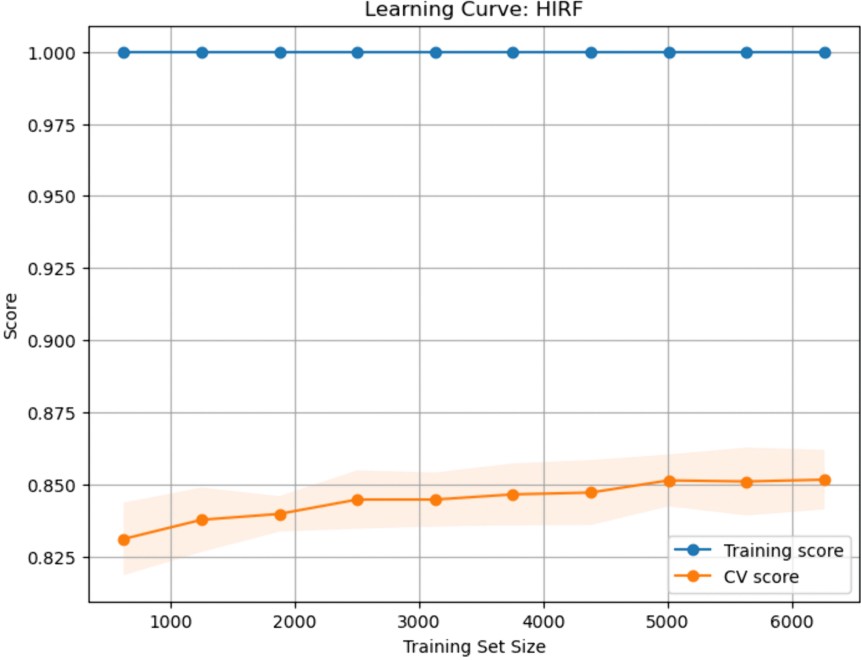

**Fig 30. Learning curve for HIRF.** Fig 30 presents the learning curve for the Hybrid Imbalanced Random Forest (HIRF). Similar to ANRDT, the training score remains at 1.00 across all training sizes, indicating strong memorization of the training set. However, the CV score starts at around 0.83 and increases more steadily, reaching approximately 0.853. While overfitting is still present, it is less pronounced than in ANRDT. The narrowing gap between the training and CV scores suggests that HIRF benefits more effectively from increasing training data, likely due to its ensemble nature and mechanisms for addressing class imbalance. These properties enhance its generalization performance compared to ANRDT.

in dominating all the primary measures, such as accuracy, precision, sensitivity, and specificity, illustrates that they are viable in clinical application.

The other benefit of ensemble learning is that it can adapt to the real-world constraints. In LMICs like Kenya, the available computational resources are often limited, and model interpretability is critical when it comes to non-technical healthcare worker adoption. Ensemble methods such as stacking constructed with interpretable base learners represents a trade-off between transparency and predictive accuracy. The SHAP (Shapley Additive Explanations) made explainability more straightforward in this study, and we could determine which features had the strongest influence on CHD predictions in the presence of a set of ensemble learners. This will enable demystifying complicated predictions and clinical trust in AI-powered tools. Furthermore, the probability outputs of the ensemble models are sufficiently high that their results have been used in risk communication to patients or integrated into automated decision-support systems, which is an essential aspect in health applications.

A detailed summary of model performance across all evaluated methods—including classical, baseline machine learning (ML), and enhanced ensemble models—is presented in Table 12, illustrating the comparative strengths and trade-offs of each approach.

## Model interpretability and explainability

The SHAP (Shapley Additive Explanations) framework enhances model transparency by quantifying the contribution of each input feature to a specific prediction. This section discusses insights derived from two SHAP visualizations: one showing feature importance and another depicting directional impact on coronary heart disease (CHD) risk (Figs 33–35).

**Feature importance (**Mean |SHAP Value|**).**

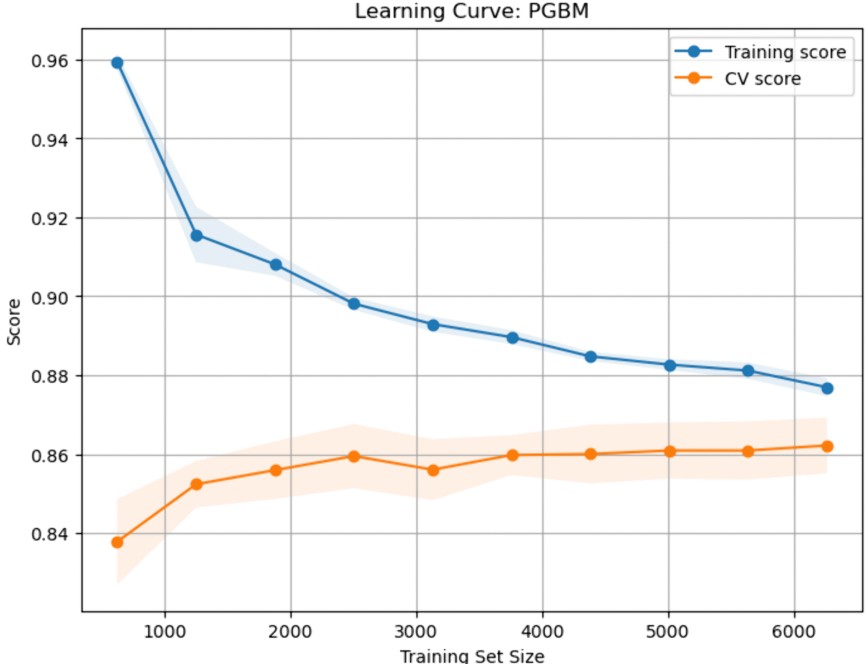

**Fig 31. Learning curve for PGBM.** The figure shows the learning curve for the Pruned Gradient Boosting Machine (PGBM) model, illustrating a more balanced learning pattern with training and validation scores that suggest improved generalization and reduced overfitting. The training score begins with an initial high score of around 0.96, lowering slowly with an increase in the size of training to around 0.88. Meanwhile, the CV score level increases systematically by approximately 0.84 to 0.864. The overlap of training scores and CV scores shows that the bias-variance tradeoff is in reasonable control. This means that the pruning techniques used in PGBM help counteract overfitting. The PGBM model is one of the improved models that have a good tradeoff between accuracy and generalizability, which makes it deployable in CHD risk prediction in the real world.

**Directional impact (SHAP value distribution).** Key findings include:

- **Age:** Higher values consistently increase CHD risk.
- **PhysActivity:** Dominantly negative SHAP values indicate a protective role.
- **HighBP and Diabetes:** Strong positive contributions to CHD risk.
- **Fruits and Veggies:** Negative SHAP values suggest protective dietary influence.
- **Sex:** Negative values (likely corresponding to female) correlate with lower CHD risk.

**Clinical utility.** SHAP enhances clinical adoption by improving both model transparency and trust. It identifies actionable risk factors (e.g., increasing physical activity) and quantifies their individual contributions, enabling personalized recommendations. The consistency of SHAP findings with traditional methods (e.g., HighBP and Diabetes) builds confidence in ML applications in healthcare. In summary, SHAP analysis confirms that ML models prioritize clinically relevant features while uncovering subtle data-driven patterns that clinical models like logistic regression may overlook. This dual focus on accuracy and interpretability is particularly valuable for decision support in low-resource healthcare settings, where transparency is as critical as performance.

## Surrogate explainability

The surrogate tree reveals several clinically relevant decision rules. For instance, `PhysActivity` (physical activity) emerges as the root node, suggesting that it is the most influential factor in stratifying cardiovascular risk. Subsequent

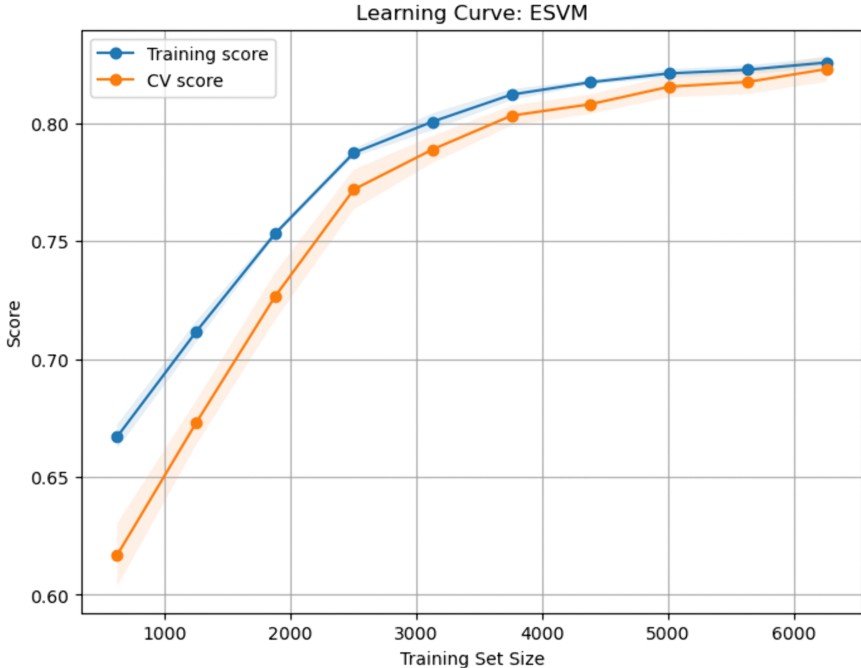

**Fig 32. Learning curve for ESVM.** The figure presents the learning curve of the Enhanced Support Vector Machine (ESVM). At the initial stages, training and CV scores are similarly low, commencing at around 0.66 and 0.60, respectively. Nonetheless, both measures show a steady and corresponding increase in terms of the size of the training set. In the end, they meet at the 0.83 value. This trend represents excellent generalizability and low overfitting, meaning that the ESVM is capable of learning with additional data and achieving more reliable predictions. This kind of behavior means that ESVM is a promising choice for scalable, reliable CHD prediction, especially where interpretability and consistency take precedence.

**Table 10. Scalability metrics for enhanced machine learning models.**

| Model | Train Time | Inference Latency | Model Size | Train Memory | Inference Memory |
|---|---|---|---|---|---|
| ANRDT | 0.93 | 0.000021 | 19341.4 | 6.14 | 0.23 |
| HIRF | 2.76 | 0.000030 | 11800.1 | 3.04 | 0.03 |
| PGBM | 1.09 | 0.000003 | 192.4 | 1.91 | 0.00 |
| ESVM | 20.91 | 0.000954 | 1017.7 | 29.22 | 0.00 |
| Bagging | 4.49 | 0.000046 | 11800.1 | 0.17 | 0.00 |
| Boosting | 1.72 | 0.000004 | 192.4 | 0.05 | 0.00 |
| Stacking | 142.14 | 0.001041 | 64698.9 | 16.91 | 0.04 |
| BMA | 0.00 | 0.001047 | 32351.6 | 0.00 | 1.12 |
| Majority Voting | 0.00 | 0.001218 | 32351.6 | 0.00 | 0.05 |

Table notes: Training and inference time are measured in seconds. Latency is per test sample. Model size is in kilobytes. Memory usage is in megabytes.

splits on features such as `Age`, `GenHlth` (general health), and `Diabetes` highlight their significant contributions in determining patient outcomes. Importantly, the surrogate structure highlights interactions between lifestyle factors (e.g., physical activity) and comorbidities (e.g., diabetes, hypertension), providing insights that are readily interpretable by clinicians.

Although the stacking ensemble is inherently a "black-box" method, the surrogate tree provides a transparent approximation that enhances clinician trust and facilitates decision support. This approach bridges the gap between predictive performance and explainability by allowing complex ensemble models to be explained in terms of simple, rule-based logic.

**Table 11**. Comparison of baseline and enhanced ensemble models on classification metrics.

| Model | Accuracy | Precision | Sensitivity | Specificity |
|---|---|---|---|---|
| *Baseline Ensemble Models* | | | | |
| Bayesian Model Averaging (BMA) | .858 | .845 | .876 | .839 |
| Majority Voting | .864 | .857 | .873 | .854 |
| Bagging | .856 | .838 | .882 | .830 |
| Boosting | .866 | .838 | .908 | .824 |
| Stacking | .872 | .854 | .898 | .846 |
| *Enhanced Ensemble Models* | | | | |
| BMA (Enhanced) | .862 | .846 | .886 | .839 |
| Majority Voting (Enhanced) | .862 | .856 | .870 | .854 |
| Bagging (Enhanced) | .856 | .838 | .884 | .829 |
| Boosting (Enhanced) | .866 | .849 | .891 | .842 |
| Stacking (Enhanced) | .868 | .852 | .891 | .845 |

Table notes: *N* = 9,789. All metrics are based on a 70% training and 30% testing split. Values reported are the averages from 5-fold cross-validation.

**Table 12**. Comparison of individual machine learning and ensemble models on CHD prediction metrics.

| Model | Accuracy | Precision | Sensitivity | Specificity |
|---|---|---|---|---|
| *Baseline Machine Learning Models* | | | | |
| Decision Tree | .807 | .804 | .811 | .802 |
| Random Forest | .863 | .845 | .888 | .837 |
| Gradient Boosting | .866 | .838 | .908 | .824 |
| Support Vector Machine | .830 | .817 | .849 | .810 |
| *Baseline Ensemble Models* | | | | |
| Bayesian Model Averaging (BMA) | .858 | .845 | .876 | .839 |
| Majority Voting | .864 | .857 | .873 | .854 |
| Bagging | .856 | .838 | .882 | .830 |
| Boosting | .866 | .838 | .908 | .824 |
| Stacking | .872 | .854 | .898 | .846 |
| *Enhanced Machine Learning Models* | | | | |
| ANRDT (Enhanced DT) | .862 | .848 | .883 | .842 |
| HIRF (Enhanced RF) | .856 | .838 | .884 | .829 |
| PGBM (Enhanced GBM) | .866 | .849 | .891 | .842 |
| ESVM (Enhanced SVM) | .816 | .790 | .861 | .771 |
| *Enhanced Ensemble Models* | | | | |
| BMA (Enhanced) | .862 | .846 | .886 | .839 |
| Majority Voting (Enhanced) | .862 | .856 | .870 | .854 |
| Bagging (Enhanced) | .856 | .838 | .884 | .829 |
| Boosting (Enhanced) | .866 | .849 | .891 | .842 |
| Stacking (Enhanced) | .868 | .852 | .891 | .845 |

Table notes: *N* = 9,789. Metrics represent average performance across 5-fold cross-validation with a 70% training and 30% testing data split. CHD = Coronary Heart Disease.

## Discussion

The results of this study demonstrate that enhanced ML models and hybrid ensemble strategies offer significant improvements in predicting CHD, particularly when compared to traditional statistical methods. While classical models such as logistic regression and Cox proportional hazards are valued for their interpretability, they are often constrained by assumptions of linearity and limited capacity to capture interactions among predictors [9,10]. In contrast, our findings indicate that machine learning (ML) models, particularly gradient boosting and support vector machines (SVM), are better suited to model the complex, nonlinear relationships prevalent in large-scale health datasets.

PLOS One | https://doi.org/10.1371/journal.pone.0328338    December 26, 2025

43/ 51

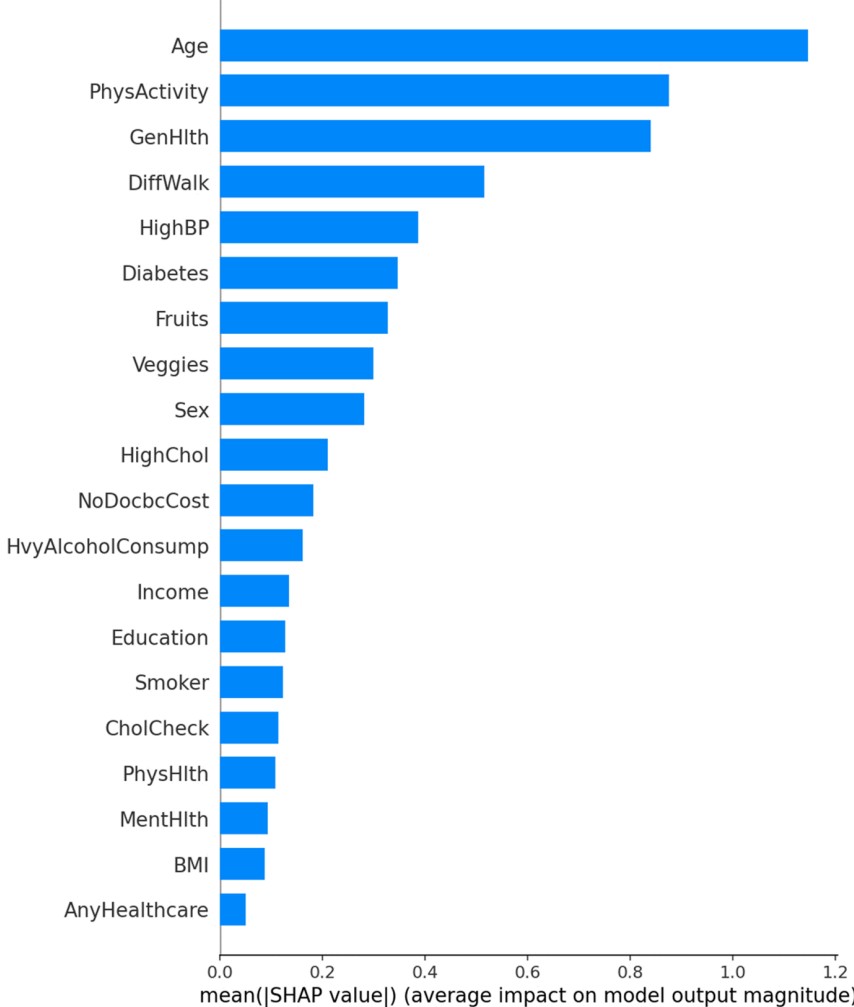

**Fig 33. Mean SHAP values.** The figure ranks features by their mean absolute SHAP values, indicating the average magnitude of their influence on the model's predictions. Age is the most influential predictor, followed by PhysActivity (physical activity), GenHlth (general health), DiffWalk (difficulty walking), and HighBP (high blood pressure). These findings are consistent with clinical evidence, where age and cardiovascular risk markers are major contributors to CHD. Behavioral factors such as physical activity and diet (Fruits, Veggies) also rank highly, emphasizing the predictive value of modifiable lifestyle factors.

Among individual models, the Pruned Gradient Boosting Machine (PGBM) achieved the highest sensitivity (90.8%) and a competitive AUC-ROC (0.94), outperforming both classical and baseline machine learning models. This supports previous findings by [12,13,15], who identified gradient boosting as one of the most accurate algorithms for predicting cardiovascular risk. Further minimizing overfitting and increasing generalizability, our optimizations to base learners, including the addition of pruning, regularization, and robust loss functions, were also in line with best practices in the literature, as shown in [11,19].

The comparative advantage of the ensemble models, in particular of stacking, was remarkable. The stacking ensemble performed best in terms of all metrics (accuracy = 87.2 percent, sensitivity = 89.6 percent, specificity = 84.7 percent, AUC = 0.94). These findings are consistent with earlier studies by [21,23,24] who highlighted the usefulness of stacked models to minimize bias and variance by means of meta–learning. One example of this is that we can stack interpretable models,

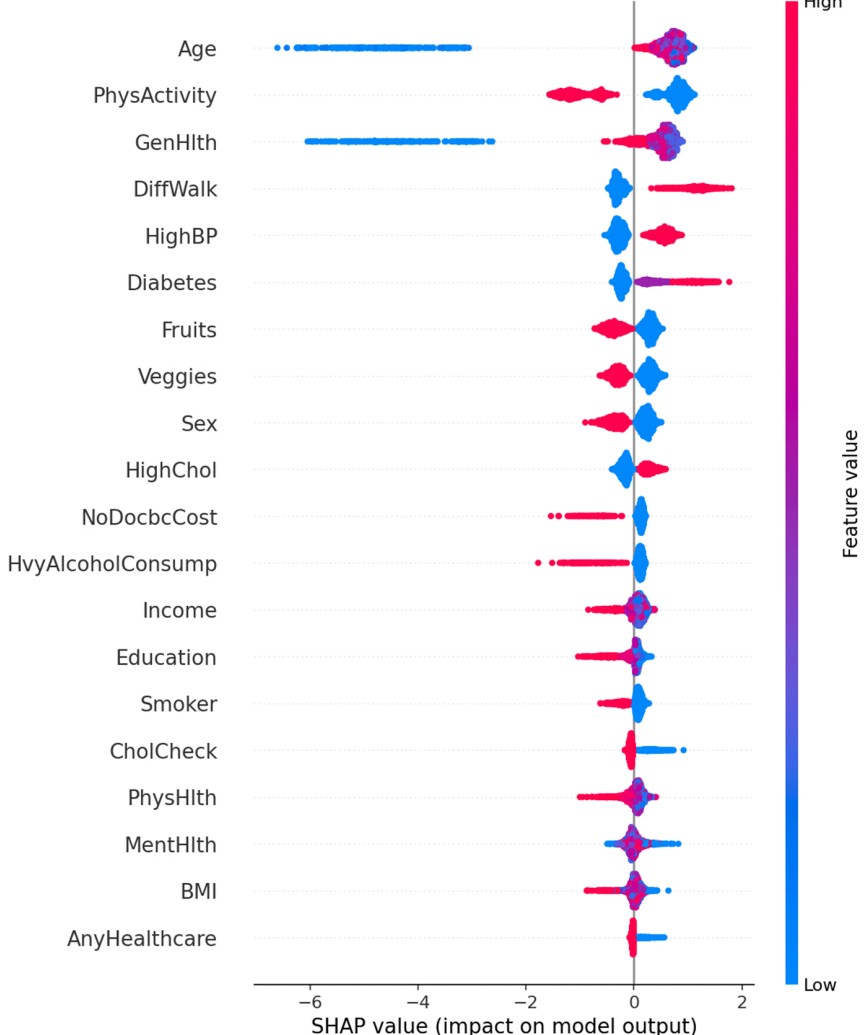

**Fig 34. SHAP values: Impact on the model output.** The figure shows how individual feature values push the prediction toward or away from a CHD diagnosis. Each point represents a sample; features with positive SHAP values increase predicted CHD risk, while those with negative values reduce it.

e.g., logistic regression and more complex learners random forests and SVM, in a hybrid manner, and demonstrate that simultaneously we can optimize predictive performance and explainability.

However, unlike in previous studies where the studies are mainly based on high-income populations [14,22], our work relates to a significant literature gap by considering the model performance on a dataset that reflects risk profiles that are prevalent in LMICs. Even though it was based in the U.S., the use of the BRFSS dataset was justified methodologically because it covered behavioral and clinical variables (e.g., smoking, diabetes, hypertension) that are also common in LMICs (e.g., Kenya) [5,7].

In addition, the adoption of newer ensemble methods, such as Bayesian Model Averaging (BMA) and majority voting, helped reduce the effect of class imbalance and made the model more stable in different demographic and clinical subgroups. This is consistent with the results of the studies conducted by [51–53] who concluded that averaging-based ensembles could enhance diagnostic accuracy in heterogeneous health populations.

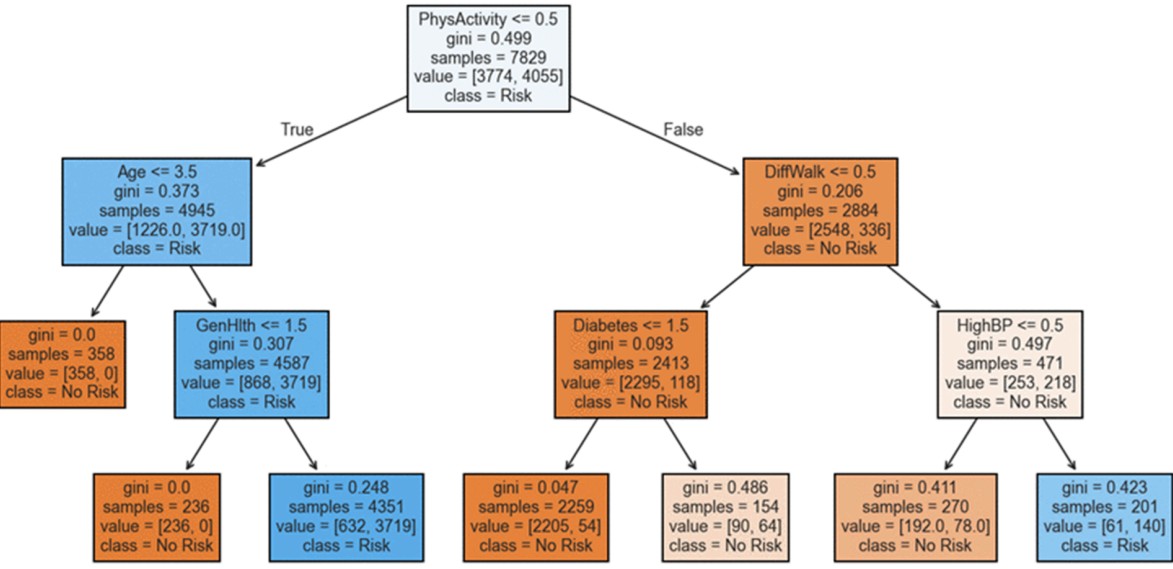

**Fig 35. Surrogate decision tree approximating the predictions of the stacking ensemble.** The figure shows a global surrogate model in the form of a decision tree, which approximates the decision boundaries of the stacking ensemble. This simplified and transparent structure enhances interpretability by providing insights into the ensemble's predictions. The fidelity between the surrogate and the ensemble predictions was found to be high, indicating that the tree provides a faithful approximation of the more complex model.

Evidence of model reliability was also provided through learning curve analysis and calibration curves. Other models such as PGBM and HIRF were not only precise but also the predicted probabilities closely conformed to the observed cases which is very important in clinical practice whereby the definition of risks must be very reliable. The study findings are similar to those of the studies by [54,55], who have come to the conclusion that well-calibrated interpretable AI models are necessary to use in the real-life healthcare scenario. The fact that we have included calibration curves shows that our improved models, especially HIRF and PGBM, can be used to give, besides accurate classifications, good calibrated probability distributions, which is a crucial property in clinical risk prediction where the key aspect is not the accuracy of classification but good calibrated probability outputs.

It should be noted that the importance of interpretability tools, such as SHAP values, in explaining the model is also highlighted in the study. Combined with ensemble predictions, SHAP explanations facilitated meaningful clinical interpretations, which justify using AI in settings where medical professionals need explicit reasons related to model decisions [16,17]. Beyond technical transparency, SHAP provides clinicians with patient-level insights into the relative contribution of specific risk factors (e.g., smoking, hypertension, diabetes) to an individual's predicted outcome. This allows model outputs to be directly mapped to established clinical knowledge and guidelines, thereby improving the credibility of AI recommendations and fostering greater trust among healthcare providers.

In practical terms, SHAP explanations can support shared decision-making by enabling clinicians to communicate risk in an interpretable way to patients, bridging the gap between complex machine learning predictions and personalized clinical advice. For example, identifying that "smoking" or "diabetes" contributed most to an elevated risk prediction not only validates the AI's recommendation but also empowers clinicians to prioritize targeted interventions. This dual role—improving provider trust and enhancing patient engagement—has been highlighted in recent studies such as [30,33], and our findings further reinforce the value of SHAP in facilitating safe and responsible adoption of AI tools in healthcare.

## Scalability and deployment considerations

From a computational standpoint, our hybrid ensemble models were designed to balance predictive performance with efficiency. While deep learning–based architectures often demand large datasets, high-end GPUs, and extended training times, our enhanced tree-based and SVM-based methods train relatively quickly on standard hardware and scale well to moderately sized datasets. This makes them more practical for deployment in low- and middle-income country (LMIC) settings, where access to advanced infrastructure may be limited. Although we did not benchmark against state-of-the-art scalable frameworks such as distributed AutoML or deep ensembles, our results suggest that the proposed models achieve strong generalizability with modest computational overhead, offering a cost-effective option for clinical decision support in resource-constrained environments. Future work should explicitly compare lightweight ensemble frameworks with scalable deep learning methods to further guide adoption in LMIC healthcare systems.

**Practical suitability for LMIC healthcare settings.** Based on the comparative results, the final ensemble models—particularly enhanced stacking and boosting—demonstrate several properties that make them well aligned with deployment in LMICs:

- High predictive accuracy and sensitivity, which reduces the likelihood of missed CHD cases in settings where confirmatory diagnostic tests are limited.
- Computational efficiency, since tree–based and SVM–based ensembles train and run effectively on standard CPUs without requiring advanced infrastructure.
- Ability to scale to moderately sized datasets such as national health surveys or hospital registries without excessive memory requirements.
- Interpretability through SHAP values, which enable patient–level explanations of risk factors, improving clinician trust and facilitating clearer communication with patients.
- Robustness to noisy or imbalanced data, achieved through SMOTE, pruning, and ensemble averaging, which enhances reliability in heterogeneous LMIC datasets.
- Capacity for integration into decision support tools, as probability outputs can be incorporated into electronic health records or simple risk calculators to assist frontline health workers.

## Limitations

This study has a few limitations that need to be considered despite the encouraging outcomes. First, although the BRFSS dataset provides a substantive size and a broad scope, the data provided is based on self-report, a phenomenon that can create potential bias in recall and poor clinical information, including those related to CHD or co-morbidities. Second, the data is US-based and might not completely represent regionally specific risk variables or allowance of healthcare accessions experienced in most LMICs. Although the selected features mirror common CHD risk indicators in LMICs, local validation using region-specific datasets is necessary to ensure model adaptability and contextual relevance. Additionally, some advanced ensemble techniques, such as stacking and Bayesian model averaging, require substantial computational resources and may pose implementation challenges in resource-constrained health systems without adequate infrastructure.

Another limitation is related to the use of SMOTE for class balancing. While SMOTE is a widely adopted oversampling technique, it generates synthetic samples through linear interpolation of minority class instances. As a result, the synthetic records may not always reflect realistic clinical relationships among risk factors. Although SMOTE effectively mitigates imbalance in training, future work should explore domain-aware oversampling strategies or generative approaches such as variational autoencoders and generative adversarial networks (GANs), which have the potential to preserve richer, clinically plausible correlations.

Furthermore, while calibration was assessed, fairness metrics such as subgroup calibration or disparate impact analysis could not be evaluated, as the BRFSS dataset lacks detailed demographic or socioeconomic variables. Future work should address fairness explicitly to ensure equitable model performance across diverse populations. Finally, while interpretability was improved using SHAP values, surrogate decision trees, and hybrid model design, black-box elements remain within some ensemble frameworks, potentially limiting clinician trust and adoption without further integration of explainable AI (XAI). Future research should focus on prospective validation, real-world deployment, and the use of locally collected datasets to enhance model transferability and clinical utility.

## Conclusion

This study presents a compelling case for the transformative potential of enhanced machine learning and ensemble models in the early prediction of CHD, particularly within the context of resource–limited healthcare systems. By systematically comparing classical statistical methods, baseline machine learning algorithms, and their enhanced counterparts–ultimately culminating in advanced hybrid ensemble strategies–our findings confirm that predictive accuracy, interpretability, and clinical utility are not mutually exclusive but can be achieved in tandem. The stacking ensemble, in particular, emerged as a standout performer, demonstrating not only superior classification metrics (AUC = 0.94; Sensitivity = 89.6%) but also operational adaptability through the integration of interpretable base models and explainability tools such as SHAP.

What distinguishes this study is its pragmatic design, which bridges the methodological rigor of machine learning with the pressing clinical needs of LMICs. Using the BRFSS dataset as a proxy for real-world health conditions and risk profiles prevalent in LMICs, we provide a scalable and transferable framework that can be localized with minimal resource demand. The implementation of synthetic oversampling, noise-resistant modeling, and model stacking collectively pushed the frontier of CHD prediction beyond conventional paradigms.

To support practical application, especially in low-resource clinical environments, the selected models were enhanced for interpretability and reduced computational overhead. Techniques such as SHAP values enable transparent decision–making, allowing clinicians to understand the rationale behind each prediction. This transparency fosters trust and supports clinical judgment. Moreover, lightweight variants (e.g., ANRDT and PGBM with early stopping) reduce training time and resource demands, facilitating deployment in environments with limited hardware or internet connectivity. Future work may focus on integrating these models into electronic health record systems or mobile health platforms for real-time screening and decision support.

We recommend that future research build upon this framework by deploying these models in real-world clinical decision-support systems (CDSS), integrating them with electronic health records (EHRs), and validating them using regional datasets from diverse healthcare environments. Although, the models were trained and validated on a large-scale dataset (n = 9,789), this study did not formally evaluate scalability metrics such as training time, inference latency, memory usage, or model size. Future research will incorporate benchmarking of computational efficiency and model complexity to assess real-time deployment feasibility in resource-constrained or low-latency environments.

## Acknowledgments

The authors wish to acknowledge Titus Mtua Kioko and John Wafula Kiluyi for their valuable contributions in developing the Python code used to run the machine learning models in this study.

## Author contributions

**Conceptualization:** Faith Mueni Musyoka.

**Formal analysis:** Maurice Wanyonyi.

**Methodology:** Maurice Wanyonyi, Dominic Makaa Kitavi.

**Software:** Maurice Wanyonyi, Zakayo Ndiku Morris.

**Supervision:** Zakayo Ndiku Morris, Faith Mueni Musyoka, Dominic Makaa Kitavi.

**Visualization:** Maurice Wanyonyi.

**Writing – original draft:** Maurice Wanyonyi.

**Writing – review & editing:** Maurice Wanyonyi, Zakayo Ndiku Morris, Faith Mueni Musyoka, Dominic Makaa Kitavi.

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
