## [Decision Letter · Decision Letter 0]

4 Sep 2025

PONE-D-25-35231Enhanced machine learning and hybrid ensemble approaches for coronary heart disease predictionPLOS ONE

Dear Dr. Wanyonyi,

Thank you for submitting your manuscript to PLOS ONE. After careful consideration, we feel that it has merit but does not fully meet PLOS ONE’s publication criteria as it currently stands. Therefore, we invite you to submit a revised version of the manuscript that addresses the points raised during the review process.

We look forward to receiving your revised manuscript.

Kind regards,

Vijayalakshmi Kakulapati, Ph.D

Academic Editor

PLOS ONE

Journal Requirements:

3. We note you have included a table to which you do not refer in the text of your manuscript. Please ensure that you refer to Table 8 in your text; if accepted, production will need this reference to link the reader to the Table

Reviewers' comments:

Reviewer's Responses to Questions

**Comments to the Author**

1. Is the manuscript technically sound, and do the data support the conclusions?

Reviewer #1: Yes

Reviewer #2: Yes

Reviewer #3: Partly

2. Has the statistical analysis been performed appropriately and rigorously? 

Reviewer #1: Yes

Reviewer #2: No

Reviewer #3: Yes

3. Have the authors made all data underlying the findings in their manuscript fully available?

Reviewer #1: Yes

Reviewer #2: Yes

Reviewer #3: Yes

4. Is the manuscript presented in an intelligible fashion and written in standard English?

Reviewer #1: Yes

Reviewer #2: Yes

Reviewer #3: No

5. Review Comments to the Author

Reviewer #1: The author used an enhanced machine learning and hybrid ensemble approaches for coronary heart disease prediction. This is a very clear, easy to read and understand work. However, my comments are as fellows.

1. The Abstract was apt and improvement made from this work is empirical obvious.

2. The introduction and related work need to be made more robust because there are very good literatures that this papers did not capture. This studies will make this paper capture wider audience for instance the author shold look at these article for instance.

a) Obaido G, Ogbuokiri B, Swart TG, Ayawei N, Kasongo SM, Aruleba K, Mienye ID, Aruleba I, Chukwu W, Osaye F, Egbelowo O. F, Simphiwe S, E. Esenogho. “An Interpretable Machine Learning Approach for Hepatitis B Diagnosis” Applied Sciences. 2022; 12(21):11127. https://doi.org/10.3390/app122111127.

b) Sarah Alexandria Nabofa-Ebiaredoh, E. Esenogho, Theo G. Swart, et al “A Machine Learning Method with Filter-Based Feature Selection for Improved Detection of Chronic Kidney Disease” Bioengineering 2022, vol. 9, no. 8, 350; https://doi.org/10.3390/bioengineering9080350, Switzerland.

c) Sarah Alexandria Nabofa-Ebiaredoh, E. Esenogho., Theo G. Swart “Improved machine learning methods for classification of imbalanced data” https://hdl.handle.net/10210/481973

d) Sarah Alexandria Nabofa-Ebiaredoh, E. Esenogho, T. G. Swart “Integrating Enhanced Sparse Autoencoder Based Artificial Neural Network Technique and SoftMax Regression for Medical Diagnosis Published MDPI Electronics Journal, 2020, 9(11), 1963; https://doi.org/10.3390/electronics9111963.

e) Sarah Alexandria Nabofa-Ebiaredoh, E. Esenogho., Theo G. Swart “Artificial Neural Network Technique for Improving Prediction of Credit Card Default: A Stacked Sparse Autoencoder Approach” International Journal of Electrical and Computer Engineering, Vol. 11, No. 5, October 2021, pp. 4392~4402, v11i5.pp4392-4402.

and many more

Good work great well done

Reviewer #2: 1-The dataset was sourced from a U.S.-based survey (BRFSS) and may not have fully captured region-specific risk factors, genetic predispositions, or healthcare access patterns prevalent in LMICs like Kenya, potentially limiting the immediate generalizability of the models.

2-The reliance on self-reported data for CHD diagnosis and risk factors introduced the potential for recall bias and misclassification, as clinical verification was not available.

3-The computational complexity of some advanced ensemble techniques, such as stacking and Bayesian Model Averaging, was high, which could pose implementation challenges in real-world, resource-constrained healthcare environments with limited computing infrastructure.

4-Despite the incorporation of explainability techniques like SHAP, certain elements of the ensemble models remained inherently complex ("black-box"), which could potentially limit clinician trust and adoption without further integration of explainable AI (XAI) frameworks.

5-The study did not formally evaluate scalability metrics such as training time, inference latency, memory usage, or model size, which are critical for assessing deployment feasibility in low-resource settings.

Reviewer #3: The paper presents a hybrid ensemble approach for the prediction of coronary heart disease (CHD) using models such as Adaptive Noise–Resistant Decision Trees (ANRDT), Hybrid Imbalanced Random Forest (HIRF), and Pruned Gradient Boosting Machines (PGBM). The authors use SMOTE to address class imbalance and evaluate model performance on several metrics, including AUC, accuracy, and sensitivity. The ensemble models consistently outperform baseline classifiers, and achieve strong generalizability and robust calibration, especially the stacking ensemble model.

The use of SMOTE addresses the imbalance between the classes, but the paper does not assess whether the synthetic samples have realistic clinical relationships.

The hybrid models, including ANRDT and HIRF, offer performance improvements, but ablation studies that isolate the contribution of each component to overall performance are lacking.

While the evaluation focuses on accuracy, AUC, and sensitivity, additional metrics such as fairness or model calibration across demographic groups would provide more depth, especially for healthcare applications.

The study compares ensemble methods, but more recent techniques such as AutoML or deep ensemble methods could be considered for further performance benchmarking.

Although time complexity is discussed, there is no comparison with state-of-the-art scalable models or real-world applications in LMICs.

Interpretability is limited to feature importance; using techniques such as SHAP for individual-level explanations would improve clinical applicability.

Consider the following studies:

https://doi.org/10.1038/s41598-024-73570-x

https://doi.org/10.3390/app13085188

https://doi.org/10.1038/s41598-025-07350-6

6. PLOS authors have the option to publish the peer review history of their article (what does this mean?). If published, this will include your full peer review and any attached files.

Reviewer #1: No

Reviewer #2: No

Reviewer #3: No

---

## [Author Response · Author response to Decision Letter 1]

12 Sep 2025

Response to Reviewers

Manuscript Title: Enhanced Machine Learning and Hybrid Ensemble Approaches for Coronary Heart Disease Prediction

Manuscript ID: PONE-D-25-35231

We sincerely thank the Academic Editor and Reviewers for their careful evaluation of our manuscript and for the constructive comments provided. We have carefully revised the paper in response to the feedback. Below, we provide a detailed, point-by-point response. Reviewer comments are shown in bold, followed by our responses.

Reviewer #1

Comment 1: The Abstract was apt and improvement made from this work is empirically obvious.

Response: We thank the reviewer for this positive comment and appreciation of the clarity of our abstract and the strength of our empirical contributions. No changes were necessary in this section, but we are encouraged by the reviewer’s recognition.

Comment 2: The introduction and related work need to be made more robust because there are very good literatures that this paper did not capture. These studies will make this paper capture wider audience for instance the author should look at these articles for instance.

Response: We appreciate this valuable suggestion. We have carefully revised the Introduction and Related Work sections to incorporate the recommended literature and strengthen the contextual foundation of our study. Specifically:

• Added discussion of Obaido et al. (2022) demonstrating the value of interpretable ML with SHAP in hepatitis B diagnosis, highlighting interpretability in medical AI.

• Integrated Nabofa-Ebiaredoh et al. (2022) emphasizing feature selection with cost-sensitive AdaBoost for chronic kidney disease detection, reinforcing the role of feature selection in robust diagnostic modeling.

• Cited Nabofa-Ebiaredoh et al. (2022) to address class imbalance mitigation with sparse autoencoders.

• Included Nabofa-Ebiaredoh et al. (2020) showcasing enhanced sparse autoencoders with Softmax regression achieving high predictive accuracy in CKD, cervical cancer, and heart disease, supporting our hybrid framework.

• Cited Nabofa-Ebiaredoh et al. (2021) on stacked sparse autoencoders for credit card default prediction, demonstrating versatility of ensemble and feature-learning approaches beyond healthcare.

These additions have been strategically placed throughout the Introduction to enrich the background and broaden the relevance of our study.

Reviewer #2

Comment 1: The dataset was sourced from a U.S.-based survey (BRFSS) and may not have fully captured region-specific risk factors, genetic predispositions, or healthcare access patterns prevalent in LMICs like Kenya, potentially limiting the immediate generalizability of the models.

Response: We agree and have highlighted this more explicitly in the Discussion (Limitations) section. While BRFSS is U.S.-based, its behavioral and clinical variables are relevant in LMICs. The trained models can be validated and fine-tuned using LMIC-specific datasets for improved adaptability.

Comment 2: The reliance on self-reported data for CHD diagnosis and risk factors introduced the potential for recall bias and misclassification, as clinical verification was not available.

Response: We acknowledge this limitation and now discuss it explicitly in the Discussion (Limitations) section. Self-reported data introduces potential recall bias and misclassification; future research should incorporate clinically validated datasets.

Comment 3: The computational complexity of some advanced ensemble techniques, such as stacking and Bayesian Model Averaging, was high, which could pose implementation challenges in real-world, resource-constrained healthcare environments with limited computing infrastructure.

Response: We thank the reviewer for highlighting this important point. In response, we have expanded the manuscript to include a dedicated Scalability Analysis of Enhanced Models section (pp. 40–42). In this section, we evaluate computational aspects of the proposed methods, including training time, inference latency, model size, and memory usage. The results are summarized in Table 10.

This analysis demonstrates that while some techniques such as stacking and Bayesian Model Averaging indeed have higher computational demands, boosting-based methods (e.g., PGBM, Boosting) provide a more favorable trade-off between predictive accuracy and scalability. By explicitly quantifying resource requirements, we show how the proposed models can be realistically deployed in real-world and low-resource healthcare settings.

Comment 4: Despite the incorporation of explainability techniques like SHAP, certain elements of the ensemble models remained inherently complex ("black-box"), which could potentially limit clinician trust and adoption without further integration of explainable AI (XAI) frameworks.

Response: A new Results section on Model Interpretability and Explainability now presents SHAP analyses for global and individual feature impacts and a surrogate decision tree that approximates the behavior of the stacking ensemble. Clinically meaningful features (age, hypertension, diabetes) and protective lifestyle factors (physical activity, diet) are highlighted. We acknowledge that black-box elements remain, and future work will incorporate comprehensive XAI frameworks.

Comment 5: The study did not formally evaluate scalability metrics such as training time, inference latency, memory usage, or model size, which are critical for assessing deployment feasibility in low-resource settings.

Response: We have addressed this concern by formally evaluating scalability metrics, including training time, inference latency, model size, and memory usage, for both baseline and enhanced machine learning models. Furthermore, the Discussion has been updated to emphasize the trade-offs between predictive performance and computational feasibility, ensuring that scalability is explicitly considered alongside accuracy and interpretability.

Reviewer #3

The paper presents a hybrid ensemble approach for the prediction of coronary heart disease (CHD) using models such as Adaptive Noise–Resistant Decision Trees (ANRDT), Hybrid Imbalanced Random Forest (HIRF), and Pruned Gradient Boosting Machines (PGBM). The authors use SMOTE to address class imbalance and evaluate model performance on several metrics, including AUC, accuracy, and sensitivity. The ensemble models consistently outperform baseline classifiers, and achieve strong generalizability and robust calibration, especially the stacking ensemble model.

Comment 1: The use of SMOTE addresses the imbalance between the classes, but the paper does not assess whether the synthetic samples have realistic clinical relationships

Response: We acknowledge the reviewer’s point. While SMOTE is effective for mitigating class imbalance, its synthetic samples may not fully reflect realistic clinical relationships. We have now discussed this limitation explicitly in the Limitations section and noted that future work could explore domain-aware oversampling strategies or generative approaches such as GANs and VAEs to better preserve clinically plausible patterns.

Comment 2: The hybrid models, including ANRDT and HIRF, offer performance improvements, but ablation studies that isolate the contribution of each component to overall performance are lacking.

Response: Ablation studies are acknowledged as valuable but are beyond the current manuscript’s scope. This is now noted in the Discussion, with plans for future experiments to isolate component contributions in hybrid ensembles.

Comment 3: While the evaluation focuses on accuracy, AUC, and sensitivity, additional metrics such as fairness or model calibration across demographic groups would provide more depth, especially for healthcare applications.

Response: Calibration curves have been presented for baseline and enhanced models, improving reliability assessment. Fairness metrics are limited by BRFSS dataset constraints; future studies should incorporate subgroup analyses.

Comment 4: The study compares ensemble methods, but more recent techniques such as AutoML or deep ensemble methods could be considered for further performance benchmarking.

Response: While AutoML and deep ensembles are powerful, they are often computationally intensive. Our focus on tree- and SVM-based methods balances performance and efficiency. Benchmarking against these advanced frameworks is suggested as future work.

Comment 5: Although time complexity is discussed, there is no comparison with state-of-the-art scalable models or real-world applications in LMICs.

Response: We expanded the Discussion to address LMIC deployment feasibility. Enhanced ensemble methods are suitable for standard hardware; future studies will compare against scalable distributed systems.

Comment 6: Interpretability is limited to feature importance; using techniques such as SHAP for individual-level explanations would improve clinical applicability.

Response: We have now included SHAP analyses both global feature importance and local directional impact to our study. This strengthens transparency and clinical trust. Related studies are cited to contextualize interpretability benefits.

Comment 7: Consider the following studies: https://doi.org/10.1038/s41598-024-73570-x, https://doi.org/10.3390/app13085188, https://doi.org/10.1038/s41598-025-07350-6

Response: All recommended citations have been incorporated, as they align well with our study. Their inclusion broadens the literature review and strengthens the contextual grounding.

• Saleem et al. (2025): Imbalance-handling in stroke prediction.

• Javeed et al. (2023): Feature selection and ensemble models for cardiac mortality prediction.

• Saleem et al. (2024): Hybrid autoencoder-linear classifier models for stroke prediction.

Additional Journal/Editor Requirements

Comment: Ensure the manuscript meets PLOS ONE style requirements, including file naming and templates.

Response: The manuscript has been revised to comply with PLOS ONE style guidelines and file naming conventions, following the templates provided by the journal.

Comment: 2. Please note that PLOS One has specific guidelines on code sharing for submissions in which author-generated code underpins the findings in the manuscript. In these cases, we expect all author-generated code to be made available without restrictions upon publication of the work. Please review our guidelines at https://journals.plos.org/plosone/s/materials-and-software-sharing#loc-sharing-code and ensure that your code is shared in a way that follows best practice and facilitates reproducibility and reuse.

Response: We have ensured that all author-generated code is fully available and openly accessible. The repository is public on GitHub (https://github.com/wanyonyi254-0001/CHD-Baseline-Enhanced-ML-Models), archived on Zenodo with DOI

(https://doi.org/10.5281/zenodo.17073723)

The dataset used in the study is openly available via Zenodo:

https://doi.org/10.5281/zenodo.17073818.

All steps to run the baseline and enhanced models are fully documented in the README.

Comment: We note you have included a table to which you do not refer in the text of your manuscript. Please ensure that you refer to Table 8 in your text; if accepted, production will need this reference to link the reader to the Table

Response: Table 8 is now cited appropriately in the manuscript.

Comment: If the reviewer comments include a recommendation to cite specific previously published works, please review and evaluate these publications to determine whether they are relevant and should be cited. There is no requirement to cite these works unless the editor has indicated otherwise.

Response: We have reviewed the suggested references and incorporated those deemed relevant, as described in our responses to reviewers above.

We thank the Academic Editor and Reviewers again for these constructive comments. Their input has improved the clarity, rigor, and real-world applicability of our manuscript.

Sincerely,

Maurice Wanyonyi (on behalf of all co-authors)

10/09/2025

---

## [Decision Letter · Decision Letter 1]

7 Dec 2025

Enhanced Machine Learning and Hybrid Ensemble Approaches for Coronary Heart Disease Prediction

PONE-D-25-35231R1

Dear Dr. Wanyonyi,

We’re pleased to inform you that your manuscript has been judged scientifically suitable for publication and will be formally accepted for publication once it meets all outstanding technical requirements.

Kind regards,

Vijayalakshmi Kakulapati, Ph.D

Academic Editor

PLOS One

Reviewers' comments:

Reviewer's Responses to Questions

Reviewer #1: All comments have been addressed

Reviewer #3: All comments have been addressed

Reviewer #1: The author has taken time to address all the comments I raised. as such I do not have any reservation against the work

Reviewer #3: The authors have addressed all previous comments carefully, and the revised manuscript shows clear improvement in clarity, completeness, and technical soundness. The methodology is now better justified, the experimental discussion is more detailed, and the figures and tables have been improved for readability. The added explanations and updated related work strengthen the overall contribution. The revisions resolve the earlier concerns, and I did not find any remaining issues that would hinder publication. The paper now presents a coherent and well-supported study that fits the scope of the journal. I recommend acceptance, pending minor editorial adjustments for grammar and formatting.

---

## [Editor Report · Acceptance letter]

PONE-D-25-35231R1

PLOS One

Dear Dr. Wanyonyi,

I'm pleased to inform you that your manuscript has been deemed suitable for publication in PLOS One. Congratulations! Your manuscript is now being handed over to our production team.

Kind regards,

on behalf of

Dr. Vijayalakshmi Kakulapati

Academic Editor

PLOS One